# Species invasiveness and community invasibility of North American freshwater fish fauna revealed via trait-based analysis

Guohuan Su [1,2] ✉, Adam Mertel [1], Sébastien Brosse [3] & Justin M. Calabrese[1,4,5]

While biological invasions are recognized as a major threat to global biodiversity, determining non-native species' abilities to establish in new areas (species invasiveness) and the vulnerability of those areas to invasions (community invasibility) is challenging. Here, we use trait-based analysis to profile invasive species and quantify the community invasibility for >1,800 North American freshwater fish communities. We show that, in addition to effects attributed to propagule pressure caused by human intervention, species with higher fecundity, longer lifespan and larger size tend to be more invasive. Community invasibility peaks when the functional distance among native species was high, leaving unoccupied functional space for the establishment of potential invaders. Our findings illustrate how the functional traits of non-native species determining their invasiveness, and the functional characteristics of the invaded community determining its invasibility, may be identified. Considering those two determinants together will enable better predictions of invasions.

Freshwater systems are among the most threatened ecosystems and most of the world's river basins have been severely altered by human activities[1,2]. Among them, habitat fragmentation and non-native fish introductions are the most pervasive[2]. In particular, fish introductions have markedly changed fish community structure and composition in rivers worldwide[3,4]. Nevertheless, our ability to predict invasions remains meager[5] considering both non–native species' invasiveness (i.e., the capacity of a species to colonize areas where it does not naturally belong), and native communities' invasibility (i.e., the vulnerability of native communities to non-native species establishment)[6] are poorly understood properties.

Invasiveness has frequently been assessed by comparing functional traits or life history strategies between non-native and native species from the recipient communities e.g.,[7–11]. Non-natives have been reported to belong to higher trophic levels and have distinct swimming capacities and life-history strategies compared to natives[10–12]. For instance, Olden et al.[10] revealed that established non-native fish species exhibited distinct life-history strategies compared to the native species in the Colorado River basin. Kolar and Lodge[5] also reported that established non-native fishes in the Great Lakes of North America tended to grow relatively fast, tolerate wider ranges of temperature and salinity, and have a history of invasiveness. However, few studies have tested which among these differential traits actually help a species colonize areas where it does not naturally occur. In other words, while it is clear that invasive species often feature different traits, the relationship between functional traits and species invasiveness per se (i.e., whether species invasiveness, measured as the number of watersheds where a species is recorded as non-native, increases/decreases along with trait values) remains to be explored.

[1]Center for Advanced Systems Understanding (CASUS), Helmholtz-Zentrum Dresden-Rossendorf (HZDR), Görlitz, Germany. [2]Institute of Hydrobiology, Chinese Academy of Sciences, Wuhan 430072, China. [3]Laboratoire Evolution et Diversité Biologique (EDB), Université de Toulouse, CNRS, IRD, UPS, Toulouse, France. [4]Department of Ecological Modelling, Helmholtz Centre for Environmental Research-UFZ, Leipzig, Germany. [5]Department of Biology, University of Maryland, College Park, MD, USA. ✉e-mail: guohuan.su@gmail.com

Moreover, species invasiveness likely interacts with the functional traits of recipient communities[13], which could determine community invasibility—the vulnerability to non-native species establishment—here measured as the established non-native species number in each community (i.e., the fish assemblage in each HUC8 watershed)[14–17]. Two mutually exclusive ecological hypotheses have been frequently invoked to explain community invasibility. First, the biotic acceptance hypothesis predicts that the number of successful invasive species is positively related to native species richness in the recipient community, as favorable environmental conditions sustaining high native species richness should also benefit non-native species[18,19]. In contrast, the biotic resistance hypothesis predicts a negative relationship between native and non-native species richness, because competitive interactions between native and non-native species will increase with native species richness, thus excluding most non-native species[14,20]. However, neither of these tw`o hypotheses has consistently explained non-native species richness in river basins around the world[4]. Instead, human activities, which are considered surrogates for propagule pressure and habitat disturbance, were responsible for increasing non-native species richness[4].

The lack of clear relationships between recipient community properties and invasibility might stem from the use of taxonomic diversity metrics such as richness or species identity. Those metrics may not accurately predict invasibility because the diversity in species does not predict the diversity of the functions they support[21]. In fish communities, most species are functionally redundant, whereas a few have unique functional traits[22,23]. Such functional uniqueness makes the communities and the functions they support vulnerable to environmental changes, implying that so-called ecosystem insurance[24] is only true for a few redundant functions. Thus, community invasibility might not be explained by the diversity of the functions exhibited by a community but instead by the functional redundancy among species in the focal community. Communities with functions supported by unique species should therefore be more vulnerable to invasions than communities with strong functional packing. In addition, functional relatedness between native and non-native species was also considered as another important factor that could affect

community invasibility[25,26]. For instance, Elleouet et al.[25] used a trait-based approach to show that non-native and native species filled a similar global functional space in the Mediterranean coastal marine fish fauna.

Our aim was therefore to characterize the functional structure of communities by considering the range (e.g., functional richness) and the partitioning (e.g., functional evenness, functional divergence) of functions within each community[21,22], as well as the functional relatedness between native and non-native species. Understanding whether the functional structure of local communities and functional similarity (or distinctness) between non-native and native species affect the invasion process could help identify whether community invasibility is primarily governed by biotic acceptance or by biotic resistance. Simultaneously, as human activities were considered as a non-negligible factor that facilitates the establishment of non-native species by increasing propagule pressure[4,6], we also tested the 'human activity' hypothesis at the watershed scale in our study. Thus, as the strength of one mechanism increases, the influence of the other two decreases. For instance, if propagule pressure plays a major role, we should not detect a strong pattern of functional traits and structure between native and invasive species, although invasion success can in some situations be exacerbated by the pattern of functional traits when propagule pressure is high [e.g., propagule pressure can ease invasion by providing species with traits overcoming unusual conditions (e.g., climatic) in one or a few events][27]. Instead, if functional traits and structure are more important factors, the influence of propagule pressure will be lower. In this last case, the functional similarity between non-native and native assemblages, as well as the functional structure of local communities will constitute important determinants of the invasion process.

Here, we examined how functional trait analysis can unify the species-centered and community-focused views of the invasion process and yield insights into both the invasiveness of particular species, and the invasibility of recipient communities. We used fish occurrence data from >1800 watersheds across the United States coupled with 20 fish life-history traits (i.e., morphological, behavioral) to compute two distance metrics between non-native and native species (Fig. 1), and six complementary functional diversity indices for

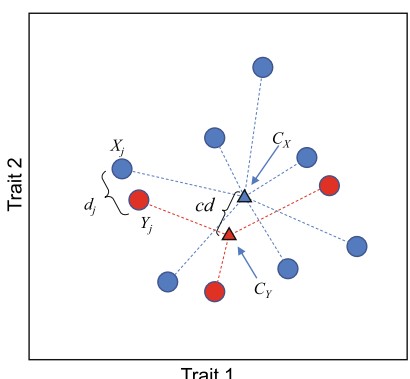

*p*: number of native species;
*q*: number of non-native species;
*n*: dimensional trait space

Two centroids:

$$C_X = [C_{Xi}] = \frac{\sum X_{ij}}{p}$$

$$C_Y = [C_{Yi}] = \frac{\sum Y_{ij}}{q}$$

Centroid distance:

$$cd = \sqrt{\sum_{i=1}^{n}(C_{Yi} - C_{Xi})^2}$$

Mean distance:

$$md = \frac{\sum_{j=1}^{q} d_j}{q} = \frac{\sum_{j=1}^{q}\sqrt{\sum_{i=1}^{n}(Y_{ij} - X_{ij})^2}}{q}$$

**Fig. 1 | An example showing how centroid distance (*cd*) and mean distance (*md*) are computed.** The *p* and *q* individual native and non-native fish species in a *n*-dimensional trait space (here *n* = 2) are represented by blue and red circles. Vector *Y_j* represents the position of non-native species *j* and vector *X_j* is the position of its nearest native species. *d_j* is the distance between *X_j* and *Y_j*. *md* is the mean distance between all non-native species and their nearest native neighbors. *C_X* and *C_Y*

(triangles) are the centroids of the *p* native species and *q* non-native species, with [*C_{Xi}*] and [*C_{Yi}*] representing the coordinates of the centroids according to all traits, i.e., [*C_{X1}, C_{X2},..., C_{Xn}*], [*C_{Y1}, C_{Y2},..., C_{Yn}*]. Here, *C_X* = [*C_{Xi}*] and *C_Y* = [*C_{Yi}*], where *C_{Xi}* and *C_{Yi}* are the mean value of trait *i* for native and non-native species. *cd* is the distance between the centroids for native (*C_X*) and non-native (*C_Y*) species.

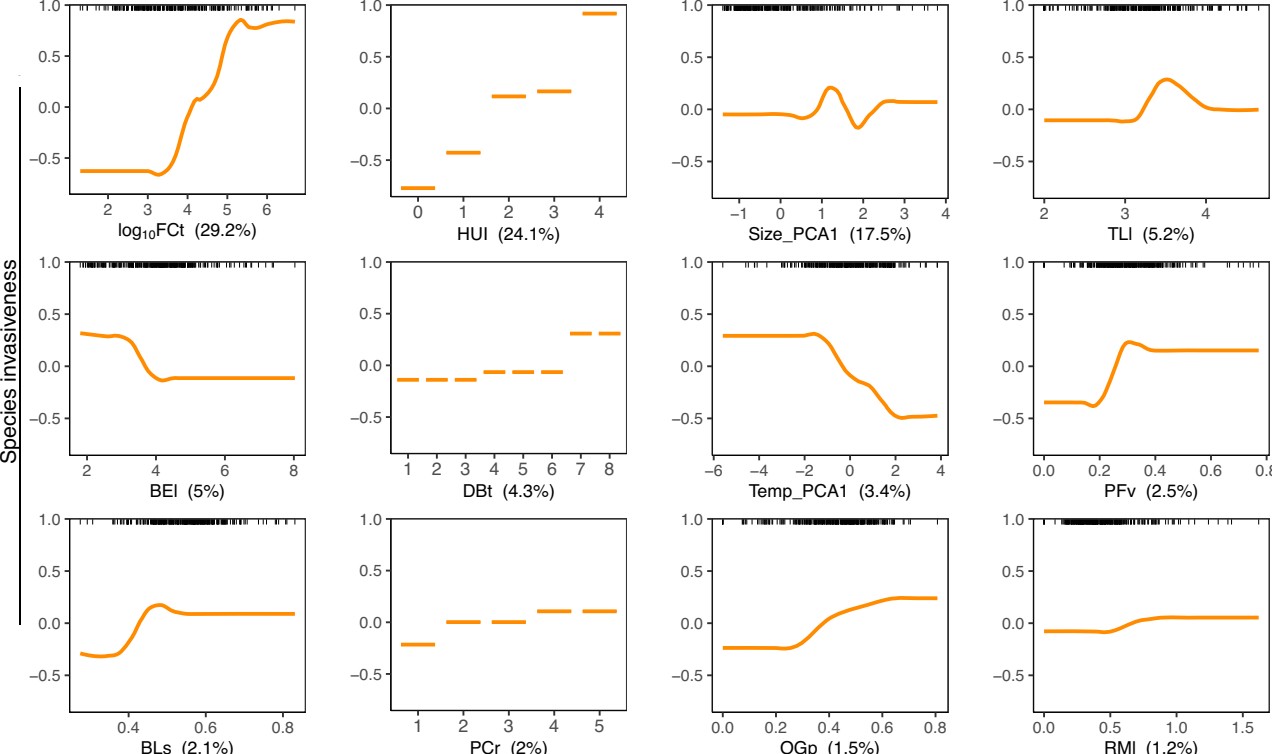

**Fig. 2 | Results of boosted regression trees showing the partial dependency between fish species invasiveness and the 12 most influencing predictors.** The value in parentheses in each panel shows the percentage of contribution of each trait considered in the model. The rugs at the top of each panel show the distribution of the species along the trait values. FCt fecundity, HUI human use index, Size_PCA1 first PCA axis of maximum body length, longevity, and mature age, BEl body elongation, DBt diet breadth, TLl trophic level, Temp_PCA1 first PCA axis of temperature range, minimum and maximum values, PFv pectoral fin vertical position, BLs body lateral shape, PCr parental care, OGp Oral gape position, RMl relative maxillary length. The figure shows the first 12 important predictors in the model, see Fig. S3 for plots of all predictors.

recipient fish communities at the watershed level. Our goal is not only to profile the functional characteristics of non-native established species, but also to quantify the vulnerability of recipient communities based on their functional attributes. We therefore expect that besides the human activities, the invasion risk posed by a non-native species results from the combination of its own functional attributes and of the functional characteristics of the recipient community. Our findings suggest that invasion success is governed by both the functional traits of non-native species that determine their invasiveness and the functional characteristics of the invaded community that determine its invasibility, and that considering these two facets together can predict species invasions more accurately.

## Results

### Functional difference between native and non-native fishes

Both translocated and exotic species showed different distributions along most of the 20 functional traits from native species. Moreover, some of the traits' average values varied gradually between native, translocated and exotic species (Fig. S1, S2). Indeed, among the 10 morphological traits, values of maximum body length and relative eye size gradually increased, while body elongation gradually decreased from native, to translocated, and to exotic species. Among the other 10 ecological and life-historical traits, average values of longevity, percent of euryhaline species, and percent of diet breadth also increased from native to translocated and to exotic species. The translocated and exotic species both showed higher fecundity than the native species (K-S test, $P < 0.001$). However, parental care for exotic species is significantly more frequent than for the native and translocated species (Chi-square test, $P < 0.001$), while the latter two groups did not differ (Fig. S1, S2).

### Species invasiveness

According to the cross-validation procedure, the boosted regression trees model (hereafter BRT) explained 42.2% of the total deviance on species invasiveness. Partial dependency plots in Fig. 2 showed that fecundity contributed the most (29.2%) to species invasiveness, followed by propagule pressure represented by a human use index (24.1%), and size-related traits (Size_PCA1, 17.5%). Trophic level, diet breadth, body elongation and temperature related traits (Temp_PCA1) had a moderate influence, each contributing about 3%-5%. The influence of the remaining 11 predictors is negligible, together contributing 11.3% (Fig. 2, Fig. S3). Generally, species invasiveness is positively correlated to fecundity, the human use index, and diet breadth, while negatively correlated to body elongation and temperature-related traits. However, none of these correlations are linear, e.g., species invasiveness does not change much at first until it increases dramatically when fecundity exceeds 1000 eggs per female (point $\log_{10}FCt = 3$, Fig. 2) then stabilizes again over 100,000 eggs per female (point $\log_{10}FCt = 5$, Fig. 2). The similar type of relationship is also found for body elongation, Temp_PCA1 and other traits. In contrast, size-related traits and trophic level showed a non-monotonic correlation to the invasiveness, with its highest values at medium-to-high values and then decreasing slightly thereafter (Fig. 2).

### Community invasibility

The BRT model explained 67.9% of the total deviance for the patterns of community invasibility. Partial dependency plots presented in Fig. 3 show the effect of a particular variable on the invasibility after accounting for the average effects of all other variables in the model. Fitted functions by the BRT model were frequently nonlinear and varied in shape. All variables related to trait-based distance, functional

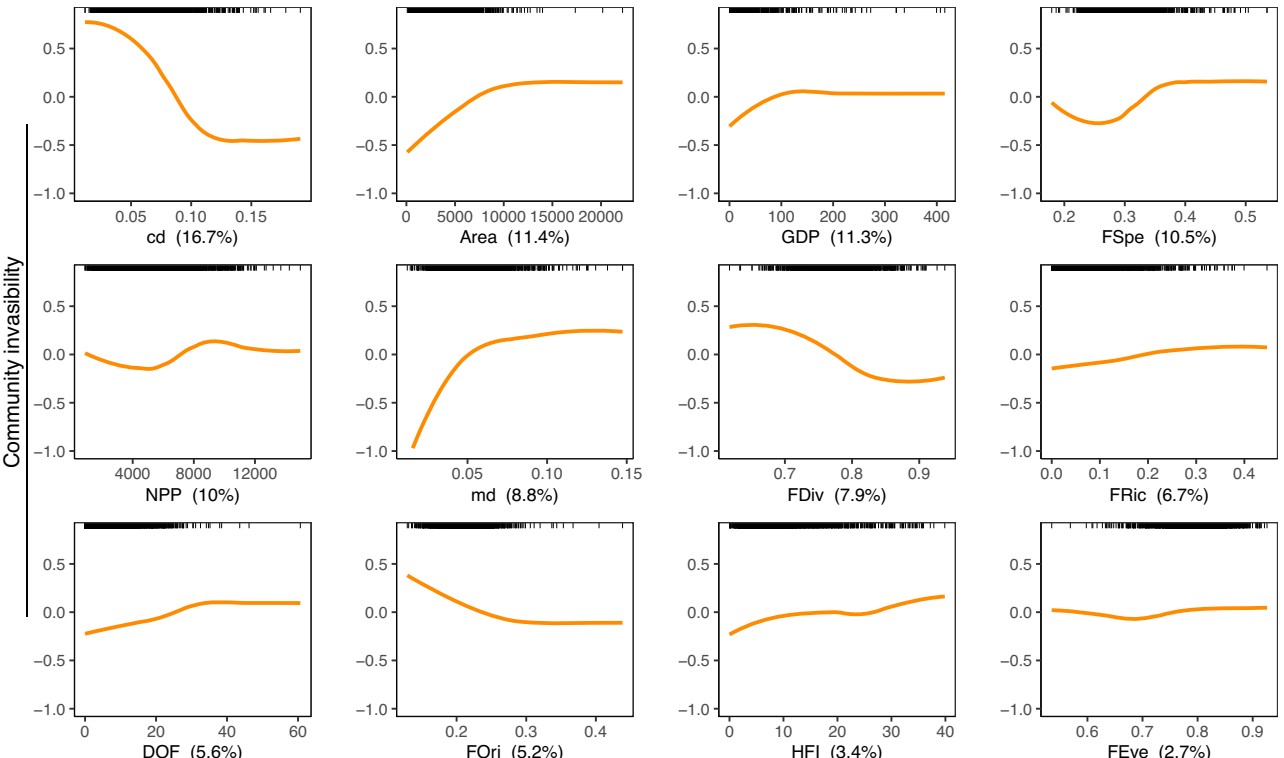

**Fig. 3 | Results of boosted regression trees showing the partial dependency between fish community invasibility and predictors related to recipient community functional structure, distance between invasive and native fish species, environment, and human activities.** The value in parentheses in each panel shows the percentage of contribution of each predictor considered in the model. The rugs in the top of each panel show the distribution of the watersheds along the predictor values. cd centroid distance, Area watershed area, GDP gross domestic product, FSpe functional specialization, NPP net primary productivity, md mean distance, FDiv functional divergence, FRic functional richness, FOri functional originality, DOF degree of river fragmentation, HFI human footprint index, FEve functional evenness.

structure, environment, and human activities were found to be correlated to community invasibility, with relative influence ranging from 16.7% to 2.7% (Fig. 3). Among them, the functional distinctness between non-native species and native fish assemblages, measured as the centroid distance between the non-native and native species assemblages, was the most influential predictor (16.7%) and was negatively related to community invasibility. Although there would be mathematical link between community invasibility and cd if native and non-native species were randomly distributed in the functional space, most traits values of invasive species are not randomly distributed in the functional space, and tend to cluster in certain regions (Figs. S1 & S2, see also case examples on Fig. 4). Thus, despite a potential mathematical link between CI and cd, the non-randomness and the clustering of functional traits of non-native species in areas of the functional space unoccupied by native species makes *cd* a main predictor of community invasibility. In contrast, mean functional distance between non-native species and their nearest neighbors (8.8%) was positively related to the community invasibility. Functional specialization (10.5%) was found to be the most influential among the five functional diversity indices and had a positive influence on community invasibility. Overall, the five functional metrics contributed 33% to the invasibility. As expected, invasibility also increased significantly with the intensity of human activities on the watershed, and the three human-related variables (GDP, DOF, HFI) contributed 20.3% to invasibility, with GDP having the greatest impact (11.3%, Fig. 3). All correlations between community invasibility and predictors were non-linear monotonically decreasing or increasing relationships. For instance, community invasibility was strongly negatively related to centroid distance between native and non-native species *(cd)* values lower than a threshold (0.12), and then not sensitive to the highest range of *cd*

values. In the same way, community invasibility increased with watershed area up to 12,000 km², but then stabilized for larger areas (Fig. 3). In addition, interactions between the 12 predictors were found in the BRT model for community invasibility, with centroid distance having strong interactions with FSpe, NPP, and FOri (Fig. S4). The three-dimensional surface plots showed that the three pairs of the most strongly interacting predictors in the BRT model have additive effects on community invasibility. The highest invasibility values were found in communities with low centroid distance between native and non-native species, high functional specialization, high available energy (NPP), and low functional originality (Fig. S4B–D).

## Discussion

Fish functional traits are different between native and non-native species. Specifically, studies considering morphological differences between natives and non-natives at the river basin scale showed that non-native species had larger size and less elongated bodies than the native counterparts[9,28,29]. Furthermore, growing evidence suggests non-native species have different traits than native species within river basins[5,10,11,30]. Our regional (watershed) approach over the continental US revealed that functional traits, including morphological, physiological, and life-historical aspects differ between native and non-native species (including translocated and exotic species), therefore paralleling previous studies at both local and global scales.

However, our findings also show that not all of the traits differing between native and non-native species contribute to the invasiveness of non-native species. Instead, species' invasiveness was only predicted by a few traits, among which fecundity and size-related traits are the most influential. High fecundity is considered as an advantage to establish and spread in a novel environment, as shown by Koehn[31] for

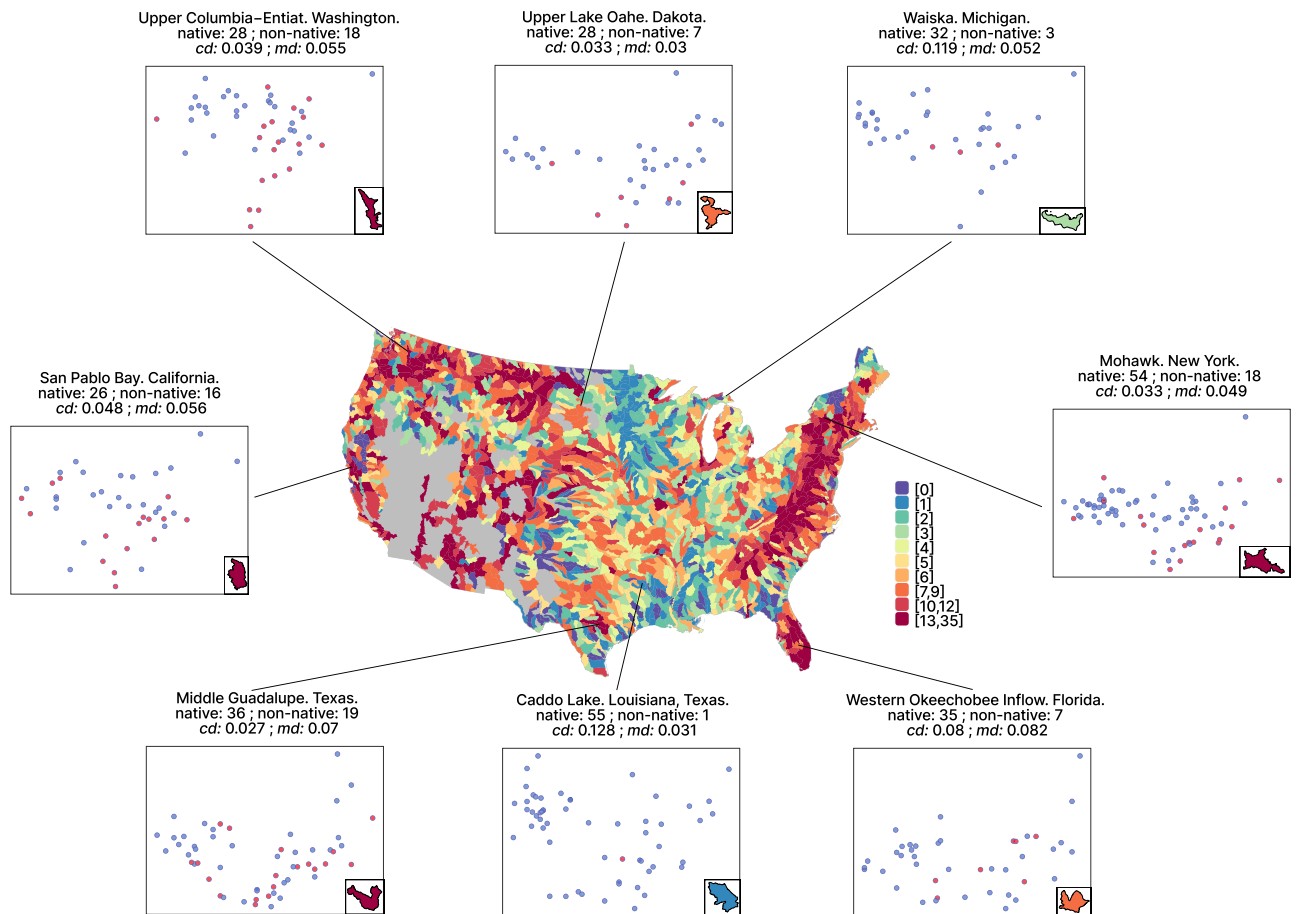

**Fig. 4 | Pattern of non-native fish species in the 1868 watersheds in the US.** The map shows the number of non-native fish species in the 1868 watersheds in the US. Scatter plots around the map show some examples of the position of native (blue dots) and non-native (red dots) species in a 2-dimensional functional space. The number of native and non-native species, and the values of the centroid distance (*cd*) and mean distance (*md*) for each watershed are indicated above each panel. Watersheds are shown on the inset map in each scatterplot.

the common carp (*Cyprinus carpio*) invasion in Australia. Moreover, life span and body size are positively linked to the dispersal ability of the species, and therefore favor the post establishment spread of large and long-lived species[8,29]. In addition, large species are often preferred for aquaculture and angling, which are among the most efficient pathways of introduction, and generate massive and widespread fish releases in natural environments[32,33] that contribute to the invasiveness of those species. Notably, our results show that the relationship between species invasiveness and functional traits is non-linear, suggesting that invasiveness is more like a threshold function of traits and determined by joint effects of several traits. Intriguingly, our results also show that top predators with large body size and relatively narrow diet breadth, are not among the best invaders. Instead, omnivorous fish with medium-to-large bodies and wide diet breadth appear to be more invasive. For instance, top predators such as great snakehead (*Channa marulius*, exotic) and Muskellunge (*Esox masquinongy*, translocated) feeding only on fishes and other aquatic animals, have been recorded to establish in 1 and 25 watersheds in the US, respectively. In contrast, the common carp (*Cyprinus carpio*, exotic) and bluegill sunfish (*Lepomis macrochirus*, translocated), with a medium-to-large body and moderate trophic level but a wide diet breadth, have been recorded to establish in 1126 and 452 watersheds in the US, respectively.

In contrast, although other traits of non-native species also significantly differed from those of native species, their contribution to species invasiveness remained negligible. This could be explained by the complexity of the invasive process (including introduction,

establishment, and spread steps, each determined by distinct drivers) and the variety of factors that might affect invasion success (e.g., propagule pressure or temporal dynamics of introduction)[26,34,35]. Indeed, our results show that the human use index, employed as a surrogate for propagule pressure in the model, remains an important predictor of species invasiveness, suggesting that human interest is a non-negligible factor in the process of species invasion, especially in the introduction stage[2]. In addition, although the distributions of most traits significantly differed between the translocated and exotic species (Table S1), the type of invasion (exotic/translocated) had limited influence on the species invasiveness model, demonstrating that species identity or native origin is a poor proxy for invasiveness compared to functional traits and human interests.

Community invasibility of US watersheds was more influenced by the functional structure of the communities than by human activities. Nevertheless, the three human-related variables collectively contributed 22.1% in explaining community invasibility, partially supporting the "human activity" hypothesis[4] and verifying the significant impact of propagule pressure. However, since we used human activity as surrogates of propagule pressure, it remains difficult to discuss the true effect of propagule pressure. We thus highlight that the level of economic activity of a given watershed (expressed by the GDP) strongly affects the community invasibility through a possible increase in propagule pressure. In contrast, human footprint and the degree of river fragmentation had surprisingly small effects on community invasibility, whereas they are often considered to be the major predictors of the number of non-native species[4,36,37]. For instance, Su

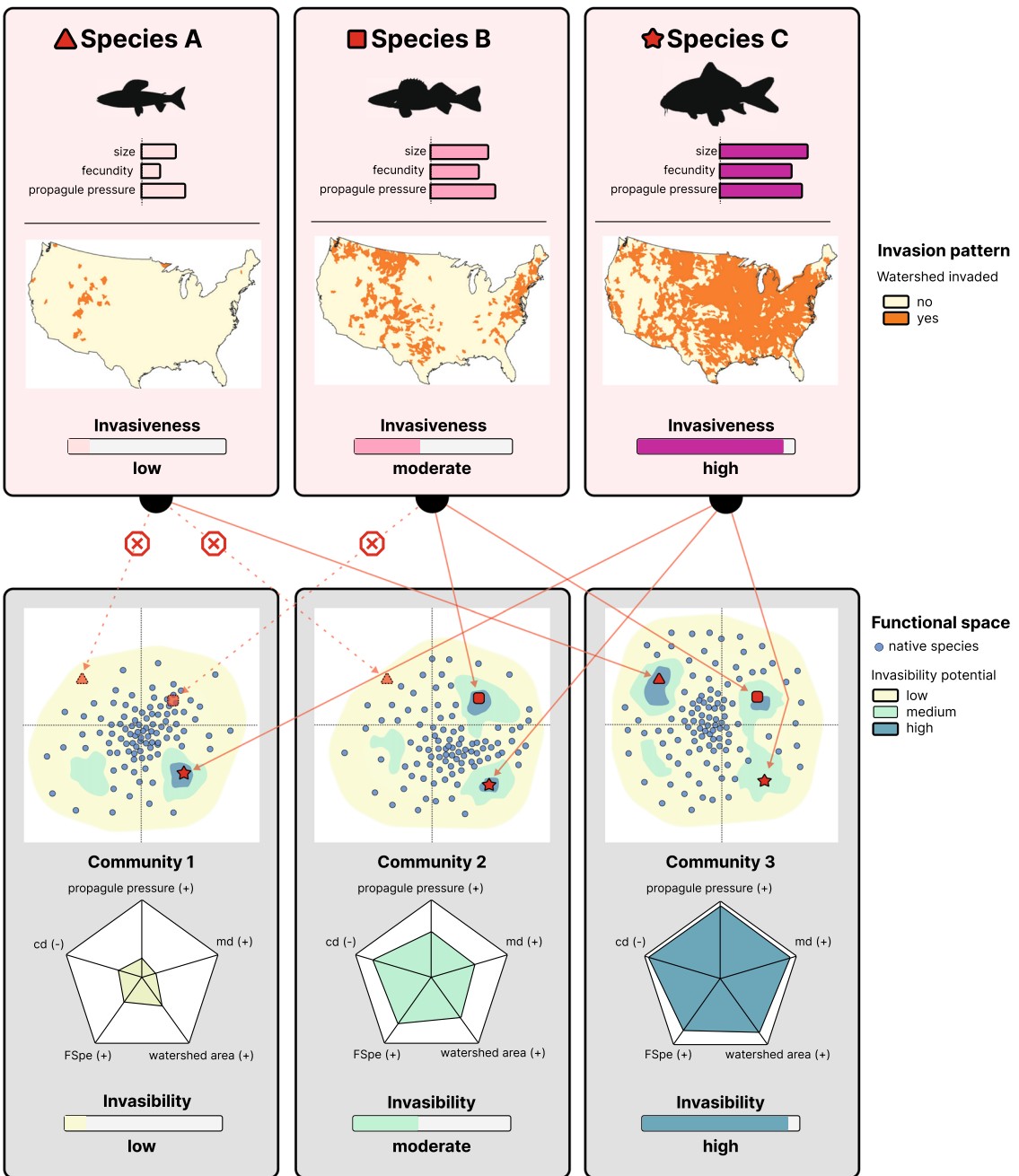

**Fig. 5 | Theoretical representation of the links between the non-native fish species invasiveness and recipient community invasibility.** Non-native species are characterized by traits promoting invasiveness and by human use for each species as a proxy for propagule pressure. Combining species traits and propagule pressure predicts species invasiveness measured as the number of watersheds where the species established (shown in orange on the maps). Species A, B and C thus have a low, moderate and high invasiveness, respectively. Meanwhile, species C also has a high chance to establish in the three recipient communities because its functional traits locate the species close to the center of the functional space of the recipient communities (i.e., low centroid distance between non-native species and native assemblage) and the non-native species traits are not redundant with those of native species (i.e., high mean distance between native and non-native species). In contrast, species A and B either located near the boundaries the local functional space (high centroid distance, species A in Communities 1 and 2) or in crowded regions of the native functional space (low mean distance, species B in Community 1) will have low chances to establish, resulting in a low invasiveness for those species. Native community invasibility is therefore determined by high establishment probability regions in the functional space containing no or few native species (darker blue areas). cd centroid distance, md mean distance, FSpe functional specialization. Note that Species A, B, and C (*Thymallus arcticus, Sander vitreus*, and *Cyprinus carpio*) are real and their invasion patterns are based on actual data.

et al.[28] revealed that patterns of change in global fish biodiversity were dominated by the introduction of non-native species in anthropized areas with high human footprint and intense river fragmentation. Our regional findings contrast with these global results as we report a relatively weak influence of human activities on community invasibility patterns across the US watersheds. This inconsistency might be rooted in the different spatial scales considered. The watershed spatial scale used here may have highlighted the role of species interactions within the communities on invasibility at the expense of some human disturbances that might be better assessed over larger spatial scales. Besides, at the watershed scale, the measurement of river fragmentation only accounts for dams located in the focal watershed, while

hydrologically mediated effects might spread far downstream[38,39]. Our findings also tend to support the biotic acceptance hypothesis, which is verified by the positive correlation between the watershed area (and thus the native species richness), NPP, and invasibility, therefore confirming that species-rich native assemblages are not insured against non-native species establishment. This was verified for watershed areas up to 12,000 km², whereas, for larger watersheds, areas are no longer linked to invasibility, which stays maximal.

More importantly, by considering the functional diversity metrics of the local community and the trait-based distance metrics between local and non-native assemblages, we were able to shed further light on the mechanisms of community invasibility. The highest invasibility was indeed recorded when native and non-native species pools from a watershed share the same functional diversity (lower centroid distance between non-native and native species assemblages than that in other watersheds), and individual native and non-native species within that assemblage are not functionally redundant (higher mean distance between the non-native species and their nearest neighbors than that in other watersheds). For instance, with similar native species number, the fish communities in watersheds with less invasive species tended to have a higher centroid distance and a lower mean distance, e.g., Caddo lake, Waiska (Fig. 4). In contrast, the communities in Upper Columbia-Entiat, San Pablo Bay, Middle Guadalupe, and Mohawk with more invasive species tended to have a lower centroid distance and a higher mean distance (Fig. 4). This indicates consistency in the pattern of most non-native species packing into the center of the native species' functional space, but keep distance from their native neighbors, which might avoid competitive interactions that are prone to reduce the chances of establishment. Although competitive interactions cannot be properly tested here, field studies on specific communities to test this hypothesis are warranted. Nevertheless, the observed process can be viewed as an environmental filtering effect that increases the overall functional similarity between native and non-native species pools in the same watershed (thus resulting in an apparent biotic acceptance effect). This also explains the low and constant invasibility for higher centroid distance (*cd*) values (above a threshold of 0.12) due to non-native species holding traits falling out of the range of traits of the native community (see Fig. 5, Species A in Communities 1,2). Then, among the species that successfully passed the environmental filter, only those occupying an available functional niche can establish and not suffer from biotic resistance (see Fig. 5, Species C vs B vs A in Communities 1,2,3). The interplay between environmental filtering at the community scale and biotic resistance at the species scale, therefore, represents a joint process that may solve the long-standing debate about the environmental vs biotic determinants of biological invasions e.g.,[4,40]. It also gives weight to the doubts raised about the environmental filtering effect, which often relies on the joint effect of environment and biotic interactions[41,42]. We here show that both processes act together, but the former at the community level and the latter at the species level. Thus, communities with a lower density of species in their functional center will be the most sensitive to invasions, which is confirmed by the positive correlation between community invasibility and functional specialization. Indeed, functional specialization represents the proportion of generalist species (i.e., species close to the center of the functional space[22]) in a community, and high functional specialization indicates that more gaps are available around the functional center. In contrast, the size of the functional space (i.e., functional richness) and other metrics representing the functional structure of assemblages do not facilitate invasions. At least for the US fish fauna, functional specialization can be considered as a proxy for vulnerability to non-native species, and it could therefore be of major interest when designing management actions to avoid further invasions of the most sensitive watersheds. This is of particular importance given the current spread of non-native species throughout the world, as well as the predicted emergence of new invaders[43,44]. We thus implore future studies to evaluate the relevance of the functional specialization metric as a proxy for invasion vulnerability in other regions and on other taxa.

To conclude, our study shows the importance of functional traits in the analysis of species invasiveness and community invasibility. We confirm that functional differences between native and non-native fish species exist, but species invasiveness is dominated by only a few functional traits among them. Our results also provide insights into the mechanisms promoting community invasibility. Although our findings tend to support the human activity and biotic acceptance hypotheses, in essence, community invasibility cannot be simply driven by one of them or by their joint effect. Instead, the original mechanisms unveiled in our study suggest that functional similarity between the non-native and native species and the local community functional structure are more influential than human activity and biotic acceptance in shaping community invasibility patterns. Communities with higher levels of functional redundancy or denser functional centers would have stronger resistance to invasive species (i.e., lower invasibility). This could explain to some extent why the speciose fish communities in the Amazon river basin, which are highly redundant in functions[45,46], have received few non-native species[47]. Therefore, we suggest that prediction of biological invasion should comprehensively consider the invasiveness of non-native species and the invasibility of recipient communities, as illustrated on Fig. 5, and encourage considering the interactions between those two properties in future studies. Moreover, based on the performance of the two models, community-level properties (invasibility) that facilitate invasion appear to be more predictable than species-level properties (invasiveness), and are therefore more likely to be informative. Our study also raises a question as to why most species, whether native or non-native, tend to gather in or invade the crowded center of the functional space, even if the space near the border is almost empty. We expect that the distribution of resources and the relative position of species niches are the key factors, but further evidence is needed to confirm this.

## Methods

### Species occurrence data

Native species occurrence records (783 species considered) at the watershed scale (i.e., Hydrologic Unit Code 8, HUC8) were obtained through NatureServe (https://www.natureserve.org) and included both extant and extinct species to account for species historically present in a given watershed but extirpated as a potential consequence of various human activities such as species invasions. Occurrence records of the naturalized or established non-native fish species (300 species considered) were obtained through the U.S. Geological Survey (USGS) Non-indigenous Aquatic Species (NAS) database[48]. Non-native species data includes exotic species that were historically absent from the continental US, and translocated species that were native to the continental US but translocated to watersheds from which they were historically absent. This dataset only considered the identified species that locally create self-sustaining populations, thus we excluded records of non-self-sustaining or eradicated populations, vagrant species detected in only one sampling occasion, and non-identified or hybrid species. NatureServe and USGS data provide comprehensive species lists per watersheds, and have often been used in previous studies[49–52]. Watershed average area was 3628.7 ± 2113.2 km². This spatial grain was relevant to investigate species invasiveness and community invasibility, because native and non-native species occurring in each watershed are actually encountering each other and can thus interact. The 1868 considered watersheds cover most of the continental US, and belong to 18 river basins. Several watersheds per basin were thus considered and watersheds are thus not strictly independent units, but each account for homogeneous environmental conditions, and ecosystem structure[53]. We identified 1715 watersheds that have at least one record of a non-native species,

1556 watersheds that had translocated species (i.e., species that were native to the continental US but translocated to watersheds from which they were historically absent), and 1344 watersheds that had exotic species (i.e., species that were historically absent from the continental US). The 1868 focal watersheds contain 859 fish species, including 559 native species that have never been established in other watersheds, 233 translocated species, and 67 exotic species. Native and non-native species richness spatial patterns were contrasted, with the highest native fish richness in the watersheds belonging to the Mississippi drainage, whereas non-native species richness peaked east and west of the Mississippi (Figs. 4 and S5).

### Predictors used in the invasiveness models

**Functional traits**. We collected ten morphological traits related to fish locomotion and food acquisition and ten additional traits related to life-history and physiological functions from FISHMORPH database[54], the Fish Traits database for North American freshwater fishes[55], and FishBase[56]. The ten morphological traits are maximum body length, body elongation, relative eye size, oral gape position, relative maxillary length, vertical eye position, body lateral shape, pectoral fin vertical position, pectoral fin size, and caudal peduncle throttling. Except for maximum body length, the other nine morphological traits are relative traits that were measured as unitless ratios. The ten additional ecological traits are longevity, fecundity, mature age (i.e., age of sexual maturity), trophic level, temperature range, minimum temperature, maximum temperature, euryhaline (yes, no), parental care (non-guarders1: open substratum spawners, non-guarders2: brood hiders, guarders1: substratum choosers, guarders2: nest spawners, bearers), and diet breadth (from 1 to 9). See Table S2 for details on the 20 functional traits.

Due to insufficient information on some species, some values were missing in the raw functional trait data. Overall, 18.1% of the values were missing in the raw trait dataset of 859 fish species, and missing values distributed evenly among the native, non-native, and taxonomic orders (Fig. S6). We statistically imputed these missing values (NA) with a machine learning algorithm called 'missForest'[57,58]. This method uses a random forest trained on the observed values of a data matrix to predict the missing values and automatically calibrates the filling values by a set of iterations. In the imputation process, after each iteration the difference between the previous and the new imputed data matrix is assessed for the continuous and categorical parts, and the algorithm stops once both differences become larger[57]. It can be used to impute continuous and/or categorical data and is not biased by complex interactions or nonlinear relationships. We included the evolutionary relationships between species in the imputation process by including the first ten phylogenetic eigenvectors in the matrix to be imputed[59]. We tested the accuracy of this method in filling in missing values on a random set of 350 species with complete values for all traits. We randomly deleted 20% of the values for the 350 species, and then imputed them with 'missForest'. We then compared the simulated values to the actual values, and repeated this procedure 100 times. Finally, we quantified imputation accuracy by calculating the Spearman correlation coefficient between the actual and imputed data, which varied from 0.88 to 0.98. In contrast, the classical imputation method of filling in missing observations with the average trait values for the 17 continuous traits of the 80% species with data, produced average correlation coefficients that ranged from 0.74 to 0.88, confirming the improved performance of 'missForest' (Fig. S7).

**Human use index**. Besides the functional traits of the fish species, we applied the human use index as a surrogate in the models[29] to account for the impact of propagule pressure on species' invasiveness.

FishBase[56] provides four categorical indexes of human use, which refer to the fisheries, aquaculture, game fish and ornamental importance of each species. We assumed that each categorical index has an equivalent importance and assigned each index to 0 or 1 for each fish based on the description (e.g., 0 for never/rarely used, 1 for occasionally/commonly used). Then, we calculated the sum of the four indexes as the human use index, which varies between 0 (i.e., for species not used by human) and 4 (i.e., for species strongly used by human).

### Predictors used in the invasibility models

**Functional diversity indices**. First, we calculated trait dissimilarity between species pairs in the communities using the Gower pairwise distance[60]. This metric can handle multiple types of data (e.g., categorical, ordinal, and continuous traits). We then used principal coordinate analysis (PCoA) to build the functional space on the first five principal coordinate axes, giving rise to a five-dimensional functional space that explained over 80% of the total variance. We removed watersheds with fewer than six species to meet the criteria for calculating functional diversity indices, which resulted in 1868 watersheds for the following analyses. Then we computed six complementary functional indices that are frequently used in functional diversity studies[22,61–63]: functional richness (FRic), functional evenness (FEve), functional divergence (FDiv), functional dispersion (FDis), functional specialization (FSpe), and functional originality (FOri). The indices are briefly defined as the following: FRic—the size (i.e. convex hull) of the functional space; FEve—the regularity of traits in the functional space; FDiv—the proportion of species with the most extreme trait values; FDis—deviation of species trait values from the center of the functional space; FSpe—the mean distance of a species from the rest of the species pool; and FOri—the distance between each species and its nearest neighbor[22,64]. These six metrics were used to represent the functional size and structure (measured on the five-dimensional functional space) of the recipient fish community in each watershed.

**Trait-based distance between non-native and native species**. We computed two metrics to represent the distance between the non-native species assemblage and recipient community for each watershed in the five-dimensional functional space built up by the PCoA, which thus reflect the degree of functional redundancy between them. First, the centroid distance (*cd*) is the distance between the centroids of non-native and native species assemblages, which reflects the overall relative positions of the two groups. Second, the mean distance (*md*) is the distance between all non-native species and their nearest native neighbors, which reflects the average position of individual non-native species relative to their nearest native neighbors. Although the distance between each non-native species and the rest of the community (including natives and non-natives) could be measured, it would require considering the temporal dynamics of invasion in each watershed, which is not possible because of the lack of temporal data on non-native species settlement for most watersheds. We, therefore, considered the non-native species altogether to compute community invasibility. *md* and *cd* were computed for all 1715 watersheds with at least one non-native species. See Fig. 1 for details about how these two metrics were calculated.

**Environmental and human-related variables**. We also included the variables widely used in testing the three main hypotheses relevant to community invasibility[4]. Variables related to biotic acceptance/resistance hypotheses were selected as the native species richness (NSR), net primary productivity (NPP) and watershed area (Area, km$^2$).

NPP was taken from an online data repository (http://files.ntsg.umt.edu/), using the mean annual NPP (in g C m$^{-2}$ yr$^{-1}$) from 2000 to 2015. Area at the watershed basin scale is used as quantitative

surrogate for habitat size and heterogeneity, which is known to correlate with native freshwater fish species richness[4,65].

Because data on propagule pressure is missing for most species[26,66], it is often assessed using various proxies relating to human activities[4,52,67]. We, therefore, used the Gross Domestic Product (GDP) and human footprint index (HFI) as surrogates for propagule pressure and establishment risk in the community invasibility models. The HFI aggregates major roadways, navigable waterways, railways, crop lands, pasture lands, the built environment, light pollution, and human population density. It, therefore, includes population data, disturbance (as anthropized lands), and human and goods exchanges measured over each watershed. As a complement to GDP and HFI, we also considered the degree of river fragmentation (DOF) which measures the degree to which river networks are fragmented longitudinally by infrastructure, such as hydropower and irrigation dams[1]. This anthropic disturbance is not redundant with the GDP or HFI, and has a potential important effect on non-native species establishment[2]. In addition, these human activities may also indirectly affect invasibility by reducing the abundance or number of native species, and thus reduce biotic resistance of the native fauna and ease non-native species establishment. We, therefore, considered that the GDP, HFI, and DOF are the best available surrogates for propagule pressure and establishment risk.

HFI is a comprehensive representation of anthropogenic threats to biodiversity. The HFI dataset (resolution: 1 km²) was taken from[68], and for any grid cell, the value can range between 0–50. GDP measures the size of the economy and is defined as the market value of all final goods and services produced within a region in a given period. The GDP dataset (in US$, 1 square degree resolution) was taken from ref. [69]. DOF measures the degree to which river networks are fragmented longitudinally by infrastructure, such as hydropower and irrigation dams[1]. The DOF dataset (resolution: 500 m²) was taken from[1], and for any grid cell, the value can range between 0–100. See Table S3 for a statistical summary of the values of these variables.

If community invasibility is strongly and positively correlated to NSR, NPP, and Area, the biotic acceptance hypothesis will be supported. Otherwise, if community invasibility is strongly and negatively correlated to NSR and Area, the biotic resistance hypothesis will be supported[4]. Human activity hypothesis will be supported if community invasibility is highly correlated to the HFI, GDP, and DOF.

We mapped NPP, HFI, GDP, and DOF by their relative resolution grid data over the watershed-scale map and then calculated the mean value of all the cells covered by each watershed using QGIS version 3.18.

## Statistical analysis

We compared the distributions of 20 traits among the three assemblages (i.e. native, translocated, and exotic species) via the Kolmogorov–Smirnov test (hereafter K–S test) for continuous traits and the Chi-square test for categorical traits.

Since maximum body length, longevity, and age at maturity are highly correlated (Pearson $r > 0.7$, Fig. S8A), we used principal component analysis (PCA) and chose the first PC axis as a combined fish size trait (Size_PCA1), which represents 90.2% of the total variance (Fig. S8B). Size_PCA1 is positively related to the three original traits, indicating that a higher Size_PCA1 value means longer body length, longer longevity, and older age at maturity. Similarly, we performed a PCA for the three temperature-related traits and chose the first PC axis (Temp_PCA1), which represents 75% of the total variance (Fig. S8C). Temp_PCA1 is positively related to minimal and maximal temperature but negatively related to the temperature range, indicating that a higher value means the species is more thermophilic but has narrower temperature range. We used the

number of watersheds where a species is recorded as non-native as a proxy for its invasiveness. Then, we employed boosted regression trees (BRT) to identify which functional traits or trait combinations determine species invasiveness. We also included the type of invasion (i.e., exotic or translocated) and the human use index in the BRT model to test whether the different categories of non-native species behaved differently and control for the influence of propagule pressure. Therefore, 18 predictors were considered in this BRT model. We applied the methodology proposed by Elith et al.[70] using a BRT model that assumes a Poisson distribution of the response variable. In addition, a few species (seven out of 300 species) appeared as outliers due to extreme traits values. These traits might reflect evolutionary contingency or erroneous traits measurements, and do not follow the general patterns of the relationship between species invasiveness and fish functional traits. Those species were removed from the model analysis procedure to avoid an undue influence of those outliers, make the pattern clearer and optimize results visualization. We nevertheless compared the results before and after removing these seven species and got similar patterns, indicating that the BRT model is robust to extreme values (Fig. S9). Moreover, we log$_{10}$-transformed the fecundity trait due to the wide range of its values to better visualize the relationship in the figures. The compared results also showed that the BRT model is robust to the data transformation (Fig. S9).

For the BRT models on community invasibility, we first quantified the correlations among the above predictors for the models and found that NSR and FDis were highly correlated (Pearson $r > 0.7$, Fig. S10) with FRic and FSpe, thus we removed NSR and FDis from the following models. We then computed the established non-native species number in each of the 1868 watersheds as a proxy for community invasibility, and applied the Poisson BRT model to assess the relative importance of each of the 12 above-described predictors on the observed invasibility of the watershed-level communities.

The BRT models were fitted using the 'gbm.step' function in 'dismo' package in R[70], which allows for the specification of four main parameters: bag fraction ($bf$), learning rate ($lr$), tree complexity ($tc$) and the number of trees ($nt$). $bf$ is the proportion of samples used at each step, $lr$ is the contribution of each fitted tree to the final model, $tc$ is the number of nodes of each fitted tree determining the extent to which statistical interactions were fitted, and $nt$ represents the number of trees corresponding to the number of boosting iterations. The optimal setting of the parameters was chosen using 10-fold cross validation (CV). The procedure provides a parsimonious estimate, CV$-D^2$ (i.e., the cross validated proportion of the deviance explained), representing the expected performance of the model when fitted to new data[70]. Using CV, we explored different combinations of the parameters to be set and retained the optimal model showing the highest CV $-D^2$. Since the process includes a random or probabilistic component, to make to results reproducible and avoid chance, we reran the models 100 times under a random seed, and then calculated the mean value of relative influence of each predictor and the proportion value of the deviance explained ($D^2$). In addition, we estimated the interactions between each pair of predictors in the community invasibility model to show the interplay of the mechanisms behind them by using the "gbm.interactions" function in the "dismo" package[70].

As BRT accounts for spatial autocorrelation in neither the dependent nor predictor variables, we also ran an autoregressive error (SAR$_{error}$) model for the community invasibility patterns and compared these results with those of the BRT, to check if spatial autocorrelation affected the results. We scaled all predictor variables to have zero mean and unit variance to ensure equal weighting in the model. Quadratic terms were included in the SAR$_{error}$ model to

consider non-linear responses. We used 'poly2nb' and 'nb2listw' functions in 'spdep' R package to extract the neighbors list based on watersheds with contiguous boundaries and constructed the spatial weights matrix as the spatial constraint in the SAR$_{error}$ model. The spatial autocorrelation analysis was performed using the 'spatialreg' and 'spdep' R packages[71]. We used Nagelkerke's R-squared[72] as the pseudo R-squared to qualify the SAR$_{error}$ model's performance. After model fitting, we checked for broad spatial autocorrelation in model residuals by computing the Moran's *I* statistic[73]. The results of SAR$_{error}$ models are provided in the supplementary material (Table S4). The core drivers identified by the BRT models were confirmed by the SAR$_{error}$ analysis, suggesting spatial autocorrelation did not have an important effect on our results.

All statistical analyses were performed with R software version 4.1[74].

## Reporting summary
Further information on research design is available in the Nature Portfolio Reporting Summary linked to this article.

## Data availability
All data needed to evaluate the conclusions in the paper are present in the paper and/or the Supplementary information. FISHMORPH is publicly available through figshare (https://doi.org/10.6084/m9.figshare.14891412). Fish Traits database for North American freshwater fishes can access through https://www.sciencebase.gov/catalog/item/5a7c6e8ce4b00f54eb2318c0. Additional data and files related to this paper have been deposited in the Zenodo repository (https://doi.org/10.5281/zenodo.7802871)[75].

## Code availability
The code to reproduce all analyses and figures is available on GitHub (https://github.com/guohuansu/fish-invasion) and Zenodo (https://doi.org/10.5281/zenodo.7802871)[75].

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

## Acknowledgements

This work was partially funded by the Center of Advanced Systems Understanding (CASUS), which is financed by Germany's Federal Ministry of Education and Research (BMBF) and by the Saxon Ministry for Science, Culture and Tourism (SMWK) with tax funds on the basis of the budget approved by the Saxon State Parliament. G.S. is supported by the National Key R&D Program of China (Grant no. 2018YFD0900904). S.B. is supported by CEBA (ANR-10-LABX-25-01) and TULIP (ANR-10-LABX-0041) projects.

## Author contributions

G.S., J.M.C., and S.B. designed the study, analyzed the data, and wrote the manuscript. G.S. and A.M. collected, compiled, and visualized the data, and G.S. worked on the core data preparation and coding in R. All authors led to revising the paper and approving it for publication.

## Funding

## Competing interests

The authors declare no competing interests.
