## [Peer Review File · Nature Communications]

Reviewer comments, first round –

Reviewer #1 (Remarks to the Author):

This MS analyses species invasiveness and community invasibility of fish assemblages in US watersheds. In my opinion, the MS is well written and the analyses impressive although considerable improvement is possible.

1) The spatial dependency of the data (e.g. Fig. S4) is a clear limitation of the study that should be more clearly acknowledged. For instance, in L. 205-216 you acknowledge that you found little influence of human activities on community invasibility, in contrast to many other previous studies, and that these might be due to the scale of analyses (HUC8 watersheds).

2) Related to this you need to clarify how you computed the spatial distance for the SAR models. If it is, the geographical coordinates among HUC watersheds, this is likely not to reflect the dendritic nature of rivers and account for the spatial dependency.

3) L. 78-82. This discussion of unsaturation of communities looks naive to me. Loreau (2000 Ecology Letters 3 : 73-76) suggests "Community saturation may occur because of physical limitations, but there are no theoretical grounds for the belief that species interactions set an absolute upper limit to diversity at any scale" and I think saturation of fish communities has not been demonstrated either and that you could have native species exclusion in unsaturated communities (in fact, this is probably the case for all the known cases of native species exclusion)

4) L 123 Do increasing Temp_PCA1 axis scores mean more thermophilic species or less? It could be both and is unclear in the main text. Similar for other PCA axes. I only found the answer when I read Fig. S6.

5) Although you analyze species invasiveness and community invasibility, you do not really analyze both at the same time. This is also related to point 1. For instance, Temp_PCA1 is likely to increase invasiveness in some situations (e.g. northern US) and decrease in others (southern US).

6) L. 389 and Fig. 2. Is this Reprod_PCA1 axis really "combined reproductive trait". For me, since it includes "maximum body length" it is rather fish size (which is related to longevity, fecundity, maturity age traits, etc.)

7) L. 153-155. These interactions and Fig. S3 are barely interpreted in the MS. In Elith et al. (2008, <https://doi.org/10.1111/j.1365-2656.2008.01390.x>) I read that partial dependence plots functions "are not a perfect representation of the effects of each variable, particularly if there are strong interactions in the data or predictors are strongly correlated", which is the case. I think you should interpret the results further and more cautiously, given these results, the many intercorrelations among predictors and many other factors (e.g. hydrological alteration, upstream-downstream river gradients, etc.) not considered in the analyses. What is the meaning of the colors in Fig. S3?

8) L. 225-232. It is probably a mathematical necessity that the community mean distance is much lower than individual species distances. Can you justify that this is not the case?

9) Fig. 4 is complex but barely interpreted in the manuscript (L. 229-230) and might help much to understand and appreciate your findings.

10) L. 230-239. I think it is risky to infer such processes from your results (and do not think you identified a novel process). See <https://doi.org/10.1016/j.tree.2017.03.004> and <https://doi.org/10.1111/1365-2435.12345> for the misuses of the environmental filtering metaphor.

11) If community invasibility is the number of established non-native species in the watershed, the

relationships with functional diversity metrics such as functional richness (Fig. 3) seem also a statistical necessity. I would expect the same with cd (centroid distance), which you identified as the most important predictor: the decreasing negative relationship (first plot in Fig. 3) might be due that when you only have 2 species introduced the cd is likely by chance to be larger than when you have many species introduced. By the way, did you compute cd when you had only one species introduced? How these or HUC8 watersheds with no aliens affect the relationships?

12) L. 301 and Table S1. If you followed Brosse et al., you need to clarify that these are relative traits (ratios to remove the size effect) except for max. body length, if this is the case (?).

13) Fig. S1. I think empty density curves (only the lines) might work better (?)

14) Fig. S4. Was the Colorado River basin not included? Why? What scale did you use for the colors? For NN species, it looks strange (not log?)

15) Fig. S6. It is a bit strange to report the Spearman correlations with PCAs (which are based on Pearson correlations). Did you log-transform fecundity and MBI for the PCA? These two variables are non-linearly related (power function) and it looks strange that fecundity is more related to the second PCA axis (despite very high Spearman)

Reviewer #2 (Remarks to the Author):

What we have here is a study that attempts to assess the characteristics (traits) that make species good invaders, and the characteristics of communities that make them invulnerable, using data on alien fish species present in a large number of watersheds around North America. Such studies are not unusual, but this one claims novelty by identifying which of the traits are associated with invasiveness, rather than just which traits are associated with alien species. Perhaps such studies are unusual in fish, but they are certainly not that unusual in other taxonomic groups.

The key issues with these analyses relate to the dependent variables – species invasiveness and community invulnerability. To quote the authors, “We used the invasion frequency (i.e., the frequency of occurrence of a species in the watersheds where it is not historically present) of each established non-native species across the 1,873 watersheds as a proxy of species invasiveness.” Unfortunately, nowhere is “community invulnerability” defined, but I assume from the manuscript that it is taken to be the number of alien fish species in each watershed. Thus, species invasiveness is the number of watersheds where an alien species occurs, and community invulnerability is the number of alien species in each watershed. This is where the problems arise.

Let's take a hypothetical example. Fish species A is in 100 watersheds, but fish species B is only in 10. The authors assume that A is the better invader. But if A had been introduced to 200 watersheds, and B to 10, A has a 50% success rate and B 100%. Which is the better invader now? I would argue that some species are present in more watersheds because they have been introduced to more, not necessarily because they are better invaders. Unless you know how many watersheds each species has been introduced to, you cannot say how good an invader it is. You may well instead simply be identifying traits that make a species attractive for introduction.

That's species invasiveness, but the same applies to community invulnerability. Now some watersheds are apparently more invulnerable, but again we cannot conclude that unless we know how many species were introduced to each. The authors attempt to control for this through variables like human footprint (which they say relates to propagule pressure, but equally may relate to colonization pressure, which is probably more relevant here), but we don't know how strongly these variables relate to colonization and/or propagule pressure in this case. So we don't know how much of the variation in traits supposedly relating to community invulnerability is actually just picking up variation in colonization pressure.

In summary, the analyses presented here cannot be concluded to say anything about either species invasiveness or community invulnerability. The manuscript as presented is therefore

fundamentally flawed.

Reviewer #3 (Remarks to the Author):

General comments

A well written article that applies novel approaches to considering invasions and uncovers additional patterns of invasion using a large dataset across the continental US. The combination of analyses considering traits of invaders and invasibility of communities across the diversity of habitats and invaders provides compelling patterns related to life-history, community characteristics, etc. and sheds novel insights into how these two aspects of invasion (species-centered and (community-centered) are integrated. While the authors do a nice job of illustrating these two aspects separately, and attempt to bring them together in Figure 4 and the discussion, an improved visual demonstrating how these patterns (individual vs community) are linked would be helpful to solidify how these pieces are integrated more broadly.

Additionally, it would be helpful if the authors provided more insights into the the relationships between species invasiveness and community invasibility and the predictors. Indeed, many of these relationships appear to illustrate patterns of thresholds, yet it is unclear if this is a data-driven pattern (and rug plots would help elucidate) or are these ecological relationships. If the latter, unpacking this in the discussion, even briefly, would help direct further studies and posit additional mechanisms that affect invasions.

Finally, some specific comments are below.

Specific comments

Line 90: Inconsistent use of "nonnative" as shown here, yet "non-native" elsewhere.

Lines 125-128: The response indicates more of a ~broken stick relationship between invasiveness and reproduction traits, body elongation, etc. suggesting more of thresholds than increasing relationships across the range of values.

Figure 3 and related text. The authors should also consider the form of these relationships. For example, why do these relationships asymptote and/or demonstrate a broken-stick relationship? Why does invasibility not continue to decline when cd levels are >10? Why does invasibility not continue to increase when Area >~18,000? Are there thresholds by which invasibility changes dramatically?

While I understand this is a short-format journal, these results have important implication about which communities are the most vulnerable

Also, note the panel for Area has 6000 instead of 60000.

Lines 168-169: Given the correlations between streamflow and productivity, among other factors, this statement should be refined.

Lines 203-204: This statement should be tempered. For example, cd explained 16.8% of the variation in invasibility, yet FPT (9.1%) together with DOF (6.2%) explain 15.3%. At the same time, landscape attributes (area, NPP), together explain 23.2%. Certainly, all are having an effect, but the language should be tempered to reflect the data.

Lines 210-213: The inconsistency may also reflect the lack of propagule pressure information considered herein. See comments in methods as well, which question the links between FPT and propagule pressure as considered herein.

Lines 231-232: The statement that non-natives are "avoiding" competition is not necessarily supported by the data. There are hints at this, but the text concluding competition is a bit preliminary-especially given the challenges of demonstrating competition in the wild.

Also, despite the patterns in the inset boxes around the perimeter in Figure 4, it would be helpful to demonstrate the consistency of these patterns across watersheds to help support the text here—especially since this is a major outcome of this analysis. The authors speak to an occurrence or two and while these inset boxes do help, considering a visual (even in the Supplemental) to support the patterns between cd and md as they pertain to the ecology of invasions more broadly

would strengthen the text here. .

Lines 278-290: Given that occurrence can be affected by sampling intensities, etc., it would be helpful if the authors provided (1) a measure of how much sampling occurred in each watershed (e.g., mean, SD across watersheds or something similar); and 2) what is the temporal dataset that was included in these data? I.e., what range of dates of sampling and occurrence were included?

Also, to broaden the inference to provide the readers with the range of abiotic conditions considered herein, it would be helpful to add a summary table to the Supplemental.

Lines 313-314: On lines 295-297 the authors present a total of 862 species considered herein.

Why the discrepancy?

Lines 326-330: 20% is a high for missing data and despite the high correlations from the missForest analyses, it is hard to grasp if the taxa with missing values were those that were well outside the range of values of fishes with data. Were there any patterns as to specific taxonomic groups of invaders or native fishes without data? I.e., were these missing species closely related to those species where data occurred?

Line 351: In the previous section (~338=348), the authors present 6 functional metrics yet here consider 5-dimensional functional space?

Line 367: Area is not necessarily a measure of habitat diversity and the relationship with species richness is likely driven by species~area relationships.

Lines 366-368: Are there relationships linking any of these metrics in FPT with propagule pressure? Given the importance of propagule pressure on invasive species establishment, understanding how well this is captured in this dataset is important. If not well captured, or understood, acknowledging this may be needed. I understand this is a surrogate, but not clear how well this aspect is captured in FPT.

I can think of numerous examples where, for example, trains in the 1800s were used to transport non-native fishes across the US--which likely wouldn't be captured in these metrics for propagule pressure.

REVIEWER COMMENTS

Reviewer #1 (Remarks to the Author):

This MS analyses species invasiveness and community invasibility of fish assemblages in US watersheds. In my opinion, the MS is well written and the analyses impressive although considerable improvement is possible.

Thank you for your positive evaluation of our manuscript and for your constructive comments, which enhanced the quality of our manuscript.

Please find below our point by point response to your comments.

1) The spatial dependency of the data (e.g. Fig. S4) is a clear limitation of the study that should be more clearly acknowledged. For instance, in L. 205-216 you acknowledge that you found little influence of human activities on community invasibility, in contrast to many other previous studies, and that these might be due to the scale of analyses (HUC8 watersheds).

Yes, watersheds are not strictly independent as most river basins count several watersheds. Developing models accounting for the dendritic nature of rivers could solve this problem, but developing such model is not currently feasible (see our detailed response to point 2). We therefore acknowledged the spatial dependency as follows (lines 358-361):

“The 1,868 considered watersheds cover most of the continental US, and belong to 18 river basins. Several watersheds per basin were thus considered and watersheds are thus not strictly independent units, but each account for homogeneous environmental conditions, and ecosystem structure⁴⁹.”

The watersheds spatial grain is moreover the adequate spatial scale to consider potential biotic effects between native and non-native species that actually co-occur and potentially interact. This has been added to the methods (lines 355-358):

“Watershed average area was $3,628.7 \pm 2,113.2$ km². This spatial grain was relevant to investigate species invasiveness and community invasibility, because native and non-native species occurring in each watershed are actually encountering each other and can thus interact.”

Human activities (HFI, GDP and DOF) were here considered at the watershed scale, we can however not exclude that some distant human activities, such as dams, impact downstream watersheds. This is acknowledged in the discussion (lines 243-248):

“The watershed spatial scale used here may have highlighted the role of species interactions within the communities on invasibility at the expense of some human disturbances that might be better assessed over larger spatial scales. Besides, at the watershed scale, the measurement of river fragmentation only accounts for dams located in the focal watershed, while hydrologically mediated effects might spread far downstream^{34,35}.”

2) Related to this you need to clarify how you computed the spatial distance for the SAR models. If it is, the geographical coordinates among HUC watersheds, this is likely not to reflect the dendritic nature of rivers and account for the spatial dependency.

We used the geographic coordinates to conduct the spatial autocorrelation models and clarified how we computed the spatial distance in the methods (lines 533-535).

“We used 'poly2nb' and 'nb2listw' functions in 'spdep' R package to extract the neighbors list based on watersheds with contiguous boundaries and constructed the spatial weights matrix as the spatial constraint in the SAR_{error} model.”

We agree that this method has some constraints such as not fully capturing the dendritic nature of rivers. However, spatial autocorrelation models usually take the geographical distance as spatial constraints, and have been widely used for many studies on riverine fish communities (Leprieur et al 2011, Oberdorff et al 2019). Although models considering the dendritic structure of watersheds within rivers might perform better, but there is also a limitation that watersheds from different river basins are not connected at all, so their distance cannot be computed based on the dendritic structure as well. We could not find an applicable model in the literature to solve the cross-river basin problem, and developing and testing such a model would, by itself, necessitate a full-length research article, and is thus beyond the scope of the present effort.

Leprieur, F. *et al.* Partitioning global patterns of freshwater fish beta diversity reveals contrasting signatures of past climate changes. *Ecology letters* **14**, 325-334 (2011).

Oberdorff, T. *et al.* Unexpected fish diversity gradients in the Amazon basin. *Science advances* **5**, eaav8681 (2019).

3) L. 78-82. This discussion of unsaturation of communities looks naive to me. Loreau (2000 Ecology Letters 3 : 73-76) suggests "Community saturation may occur because of physical limitations, but there are no theoretical grounds for the belief that species interactions set an absolute upper limit to diversity at any scale" and I think saturation of fish communities has not been demonstrated either and that you could have native species exclusion in unsaturated communities (in fact, this is probably the case for all the known cases of native species exclusion)

We agree with this comment and no longer refer to the saturation/unsaturation of communities. We thus reorganized this part of the introduction and focused on biotic acceptance, biotic resistance and human activity effects (lines 77-89):

“Understanding whether the functional structure of local communities and functional similarity (or distinctness) between non-native and native species affect the invasion process could help identify whether community invasibility is primarily governed by biotic acceptance or by biotic resistance. Simultaneously, as human activities were considered as a non-negligible factor that facilitates the establishment of non-native species by increasing propagule pressure^{4,6}, we also tested the ‘human activity’ hypothesis at the watershed scale in our study. Thus, as the strength of one mechanism increases, the influence of the other two decreases. For instance, if propagule pressure plays a major role, we should not detect a strong pattern of functional traits and structure between native and invasive species. Instead, if functional traits and structure are a more important factor, the influence of propagule pressure will be lower. In this last case, the functional similarity between non-native and native assemblages, as well as the functional structure of local communities will constitute important determinants of the invasion process.”

4) L 123 Do increasing Temp_PCA1 axis scores mean more thermophilic species or less? It could be both and is unclear in the main text. Similar for other PCA axes. I only found the answer when I read Fig. S6.

Yes, Temp_PCA1 axis scores mean more thermophilic species. This is now explained in lines 485-487:

“Temp_PCA1 is positively related to minimal and maximal temperature but negatively related to the temperature range, indicating that a higher value means the species is more thermophilic but has narrower temperature range.”

We did the same for the Size PCA results (lines 482-483): “Size_PCA1 is positively related to the three original traits, indicating that a higher Size_PCA1 value means longer body length, longer longevity and older age at maturity.”

The other PCA axes were not detailed because they were not used in the BRT models, but the full results are still presented in the supplementary material (Fig S8).

5) Although you analyze species invasiveness and community invasibility, you do not really analyze both at the same time. This is also related to point 1. For instance, Temp_PCA1 is likely to increase invasiveness in some situations (e.g. northern US) and decrease in others (southern US).

Yes, the species invasiveness and community invasibility analyses were performed separately. These two analyses answered the two main questions that species with what kinds of traits are more likely to invade other regions and communities with what functional attributes are easier to invade. Separating these two questions allowed us to consider the two processes independently, but we agree they can be linked, and that’s why we also considered the functional correlation between native communities and invasive species in our models.

Going further in this direction, we could build on the present results to predict the probability that a given species will invade a focal community based on the species’ functional traits as well as the attributes of the native community and their functional correlations. Linking both processes more tightly in the manuscript would nevertheless add substantial complexity and length to the manuscript and blur the demonstration we made on the two separate questions, so we here preferred to keep the two questions separated. We nevertheless indicated in the conclusion that considering interaction between these two properties would be necessary in the future and added a theoretical figure (Fig. 5) to interpret how the functional traits can link the two aspects (lines 330-333):

“...we suggest that prediction of biological invasion should comprehensively consider the invasiveness of non-native species and the invasibility of recipient communities, as illustrated on Fig. 5, and encourage considering the interactions between those two properties in future studies.”

Fig. 5. Theoretical representation of the links between the non-native species invasiveness and recipient community invisibility. Non-native species are characterized by traits promoting invasiveness and by human use for each species as a proxy for propagule pressure. Combining species traits and propagule pressure predicts species invasiveness measured as the number of watersheds where the species established (shown in orange on the maps). Species A, B and C thus have a low, moderate and high invasiveness, respectively. Meanwhile, species C also has a high chance to establish in the three recipient communities because its functional traits locate the species close to the center of the functional space of the recipient communities (i.e., low centroid distance between non-native species and native assemblage) and the non-native species traits are not redundant with those of native species (i.e., high mean distance between native and non-native species). In contrast, species A and B either located near the boundaries the local functional space (high centroid distance, species A in Communities 1 and 2) or in crowded regions of the native functional space (low mean distance, species B in Community 1) will have low chances to establish, resulting in a low invasiveness for those species. Native community invisibility is therefore determined by high establishment probability regions in the functional space containing no or few native species (darker blue areas). cd: centroid distance; md: mean distance; FSpe: functional specialization.

6) L. 389 and Fig. 2. Is this Reprod_PCA1 axis really "combined reproductive trait". For me, since it includes "maximum body length" it is rather fish size (which is related to longevity, fecundity, maturity age traits, etc.)

Thanks for the suggestion, we agree fish size is more accurate to describe this combined trait, and we have renamed Reprod_PCA1 as Size_PCA1.

7) L. 153-155. These interactions and Fig. S3 are barely interpreted in the MS. In Elith et al. (2008, <https://doi.org/10.1111/j.1365-2656.2008.01390.x>) I read that partial dependence plots functions "are not a perfect representation of the effects of each variable, particularly if there are strong interactions in the data or predictors are strongly correlated", which is the case. I think you should interpret the results further and more cautiously, given these results, the many intercorrelations among predictors and many other factors (e.g. hydrological alteration, upstream-downstream river gradients, etc.) not considered in the analyses. What is the meaning of the colors in Fig. S3?

We agree that the partial dependence plots may not be a perfect representation, but they still provide useful basis for interpretation. We have controlled for the correlations of the predictors, and the results (Fig. S4) also showed only a few pairs of variables had relatively strong interactions. We added additional, careful interpretation of the interactions and the figure (Fig. S4). We also added 3-D surface plots for the top three strongest interacting pairs of predictors and interpreted the results in the text. The color in the figure has no specific meaning and is intended to better distinguish different interactions.

Lines 170-177: "In addition, interactions between the 12 predictors were found in the BRT model for community invasibility, with centroid distance having strong interactions with FSpe, NPP, and FOr (Fig. S4). The three-dimensional surface plots showed that the three pairs of the most strongly interacting predictors in the BRT model have additive effects on community invasibility. The highest invasibility values were found in communities with low centroid distance between native and non-native species, high functional specialization, high available energy (NPP), and low functional originality (Fig. S4 B-D)."

8) L. 225-232. It is probably a mathematical necessity that the community mean distance is much lower than individual species distances. Can you justify that this is not the case?

There is no mathematical necessity for community mean distance (cd) being lower than individual species distances (md). In fact, in most (1,137/1,715) watersheds, the cd is higher than md. Moreover, our sentence was probably unclear, because we did not compare the centroid distance to mean distance (i.e., cd vs md) but to the same index between watersheds. We rephrased this sentence to make it clearer (lines 262-267):

"The highest invasibility was indeed recorded when native and non-native species pools from a watershed share the same functional diversity (lower centroid distance between non-native and native species assemblages than that in other watersheds), and individual native and non-native species within that assemblage are not functionally redundant (higher mean distance between the non-native species and their nearest neighbors than that in other watersheds)."

9) Fig. 4 is complex but barely interpreted in the manuscript (L. 229-230) and might help much to understand and appreciate your findings.

We added more interpretation in the manuscript (lines 267-271):

“For instance, with similar native species number, the fish communities in watersheds with less invasive species tended to have a higher centroid distance and a lower mean distance, e.g. Caddo lake, Waiska (Fig. 4). In contrast, the communities in Upper Columbia-Entiat, San Pablo Bay, Middle Guadalupe, and Mohawk with more invasive species tended to have a lower centroid distance and a higher mean distance (Fig. 4).”

10) L. 230-239. I think it is risky to infer such processes from your results (and do not think you identified a novel process). See <https://doi.org/10.1016/j.tree.2017.03.004> and <https://doi.org/10.1111/1365-2435.12345> for the misuses of the environmental filtering metaphor.

Yes, you are right, although the environmental filtering effect is commonly used in the literature, it often relies on the joint effect of environment and species interactions. We have now specified that in the manuscript. We also removed the mention of a new process, but preferred to rely on the coupling of two already known processes (lines 283-288).

“The interplay between environmental filtering at the community scale and biotic resistance at the species scale therefore represents a joint process that may solve the long-standing debate about the environmental vs biotic determinants of biological invasions e.g.,^{4,36}. It also gives weight to the doubts raised about the environmental filtering effect, which often relies on the joint effect of environment and biotic interactions^{37,38}.”

11) If community invasibility is the number of established non-native species in the watershed, the relationships with functional diversity metrics such as functional richness (Fig. 3) seem also a statistical necessity. I would expect the same with cd (centroid distance), which you identified as the most important predictor: the decreasing negative relationship (first plot in Fig. 3) might be due that when you only have 2 species introduced the cd is likely by chance to be larger than when you have many species introduced. By the way, did you compute cd when you had only one species introduced? How these or HUC8 watersheds with no aliens affect the relationships?

Yes, here we used number of the non-native species as a proxy of the community invasibility, and we clarified this definition in the methods section (lines 508-509): “We then computed the established non-native species number in each of the 1,868 watersheds as a proxy for community invasibility, ...”. There is however no statistical necessity between the number of established non-native species and functional diversity metrics. Taking functional richness as an example, adding non-native species to a native community will not change the functional richness of the community if the non-native species do not possess original traits. In this case, non-native species will simply be included in the multidimensional trait space defined by the traits of the native species, without affecting the functional richness of the community.

Moreover, we here measured the functional diversity metrics for the native communities, so there is no statistical necessity between the community invasibility and the functional diversity metrics. Instead, their relationships can support either the biotic resistance or biotic acceptance hypothesis. For instance, if the community invasibility is strongly and positively correlated to the functional richness, it will tend to support the biotic acceptance, and *vice versa*.

As shown in the supplementary Fig. S1-S2, most traits values of invasive species tend to cluster either in the higher or lower values than those of native species, which means they are not randomly distributed in the functional space around the centroid. Instead, they are more likely to be on one side of the centroid or the other. Thus, we don't think there is a statistical necessity between the number of invasive species and the cd.

We computed *cd* for all watersheds with at least one invasive species (e.g., Caddo Lake shown in Fig. 4). However, we couldn't do the same for the watersheds with no aliens, because in this case, *cd* and *md* cannot be computed.

This is now indicated line 438: “*md* and *cd* were computed for all the 1,715 watersheds with at least one non-native species.”

12) L. 301 and Table S1. If you followed Brosse et al., you need to clarify that these are relative traits (ratios to remove the size effect) except for max. body length, if this is the case (?).

Yes, we clarified the sentence (lines 376-377):

“Except for maximum body length, the other nine morphological traits are relative traits that were measured as unitless ratios.”

13) Fig. S1. I think empty density curves (only the lines) might work better (?)

We modified Figure S1 and S2 as suggested.

14) Fig. S4. Was the Colorado River basin not included? Why? What scale did you use for the colors? For NN species, it looks strange (not log?)

The Colorado River basin was included. It consists of dozens of HUC8 watersheds, but some of the Colorado watersheds were not considered due to insufficient data. The red outline shows the Colorado River basin in the map below.

We used discrete and gradient colors to show the number of species in each watershed. As indicated in the legends, each color refers to the watersheds with a specific number (or range) of species. Since number of native and non-native species are quite different, in Fig.S5 we used two scales as shown on the captions. Both maps show the actual real number (not log scaled) of native or non-native species.

15) Fig. S6. It is a bit strange to report the Spearman correlations with PCAs (which are based on Pearson correlations). Did you log-transform fecundity and MBI for the PCA? These two variables are non-linearly related (power function) and it looks strange that fecundity is more related to the second PCA axis (despite very high Spearman).

Thanks for the suggestion. We changed to use Pearson correlation for the tests in our study. We tested the correlation between log-transformed fecundity and MBI, and found a strong relationship ($r = 0.79$). However, the original value of MBI is also highly linearly correlated to

LGt and MTg. Thus, conducting PCA for either original values of the four traits or log-transformed MBI and fecundity and original LGt and MTg is not applicable. It is more appropriate to keep fecundity as an original trait in the model and use the first PC axis of the PCA for MBI, LGt and MTg. See Fig. S9.

Reviewer #2 (Remarks to the Author):

What we have here is a study that attempts to assess the characteristics (traits) that make species good invaders, and the characteristics of communities that make them invulnerable, using data on alien fish species present in a large number of watersheds around North America. Such studies are not unusual, but this one claims novelty by identifying which of the traits are associated with invasiveness, rather than just which traits are associated with alien species. Perhaps such studies are unusual in fish, but they are certainly not that unusual in other taxonomic groups.

Thanks for your comments.

Yes, invasiveness has been explored for other taxa, such as plants (Colautti et al. 2014, Cadotte et al. 2018), but remains much less common for animals. To our knowledge the only study reporting invasiveness data for a species rich animal taxon is Fournier et al. (2019). This study was focused on ants, and we were not able to find similar studies focused on aquatic animals. So this aspect our study is indeed novel with respect to the aquatic animal literature

Concerning fish, several studies (the most significant of which are cited in the introduction) identified the functional profile of non-native species, but none, to date, considered invasiveness (the link between functional traits and the success of invasion, here measured as the number of localities where a species established) over a large territory such as the US. As mentioned in the introduction, we are claiming novelty for analyses of river fish, not all possible taxa.

Colautti, R. I. *et al.* Quantifying the invasiveness of species. *NeoBiota* 21: 7–27 (2014).

Cadotte, M. W., Campbell, S. E., Li, S.-p., Sodhi, D. S. & Mandrak, N. E. Preadaptation and naturalization of nonnative species: Darwin's two fundamental insights into species invasion. *Annual Review of Plant Biology* 69: 661-684 (2018).

Fournier, A., Penone, C., Pennino, M. G. & Courchamp, F. Predicting future invaders and future invasions. *Proceedings of the National Academy of Sciences*, 201803456 (2019).

The key issues with these analyses relate to the dependent variables – species invasiveness and community invulnerability. To quote the authors, “We used the invasion frequency (i.e., the frequency of occurrence of a species in the watersheds where it is not historically present) of each established non-native species across the 1,873 watersheds as a proxy of species invasiveness.” Unfortunately, nowhere is “community invulnerability” defined, but I assume from the manuscript that it is taken to be the number of alien fish species in each watershed. Thus, species invasiveness is the number of watersheds where an alien species occurs, and community invulnerability is the number of alien species in each watershed. This is where the problems arise.

Yes, we agree completely about the definitions of invasiveness and invulnerability. We have clarified these definitions in the introduction and in the methods:

Introduction:

“...while it is clear that invasive species often feature different traits, the relationship between functional traits and species invasiveness *per se* (i.e., whether species invasiveness, measured

as the number of watersheds where a species is recorded as non-native, increases/decreases along with trait values) remains to be explored.” (lines 43-46)

“...which could determine community invasibility — the vulnerability to non-native species establishment — here measured as the established non-native species number in each community.” (lines 48-50)

Methods:

“We used the number of watersheds where a species is recorded as non-native as a proxy for its invasiveness.” (lines 487-489)

“We then computed the established non-native species number in each of the 1,868 watersheds as a proxy for community invasibility, ...” (lines 508-509)

Let’s take a hypothetical example. Fish species A is in 100 watersheds, but fish species B is only in 10. The authors assume that A is the better invader. But if A had been introduced to 200 watersheds, and B to 10, A has a 50% success rate and B 100%. Which is the better invader now? I would argue that some species are present in more watersheds because they have been introduced to more, not necessarily because they are better invaders. Unless you know how many watersheds each species has been introduced to, you cannot say how good an invader it is. You may well instead simply be identifying traits that make a species attractive for introduction.

That’s species invasiveness, but the same applies to community invasibility. Now some watersheds are apparently more invadable, but again we cannot conclude that unless we know how many species were introduced to each. The authors attempt to control for this through variables like human footprint (which they say relates to propagule pressure, but equally may relate to colonization pressure, which is probably more relevant here), but we don’t know how strongly these variables relate to colonization and/or propagule pressure in this case. So we don’t know how much of the variation in traits supposedly relating to community invasibility is actually just picking up variation in colonization pressure.

In summary, the analyses presented here cannot be concluded to say anything about either species invasiveness or community invasibility. The manuscript as presented is therefore fundamentally flawed.

The central problem raised here is that we did not consider the propagule/colonization pressure and establishment success in community invasibility and species invasiveness measures. It would for sure be fantastic to be able to measure these two parameters, but it would require relevant information about failed introduction attempts. Such data are unfortunately lacking for fish, and more generally in invasion ecology. For instance, Lockwood et al. (2007) in their seminal book titled *Invasion Ecology* state that “...information on non-native populations that failed to establish is difficult to come by...” (Lockwood et al. 2007, page 85).

The same lack of data about propagule/colonization pressure, including therefore introduction attempts and failed introductions, also apply to aquatic ecosystems as frequently underlined in the literature (e.g., Garcia-Berthou 2007, Drake et al. 2015; Bernery et al. 2022). The data considered in our paper, although being among the most detailed regional data on fish invasion does not break this rule as it only contains 1,200 records of failed introduction over a total of more than 13,000 introduction events. It would therefore not be realistic to use this lacunary data to consider introduction strength or establishment rate.

We understand the reviewer's criticism, but following this logic, the lack of existing data on propagule pressure and establishment success should invalidate the results of most studies dealing with invasions (regardless of taxa), which we find unreasonable. Although we agree that using surrogates is probably not optimal, and that the results should be considered with caution (this is now underlined in the MS, see below), we still prefer using surrogates to go ahead in the understanding of the invasion mechanisms, rather than refraining from studying one of the main causes of the current global biodiversity crisis.

To cope with data deficiency, studies dealing with invasions usually consider human disturbances on recipient ecosystems and human interest for non-native candidate species as relevant proxies of propagule/colonization pressure and establishment success (Pyšek et al. 2010).

Human disturbances on recipient ecosystems are often approximated using the human footprint index (HFI), the Gross Domestic Product (GDP) or the human population density, because increasing the number of humans and economic exchanges increases the chances of introduction (Leprieur et al. 2008; Essl et al. 2011; Beaury et al. 2020, Comte et al. 2021). In addition, increasing human population and economic activity also increases environmental disturbances leading to the decline of some sensitive native species, and open niches for non-native species that are often opportunistic. This explains why the HFI and the GDP have frequently been used as proxies of propagule pressure and establishment rate (Leprieur et al. 2008, Essl et al. 2011; Beaury et al. 2020).

We therefore used the GDP and HFI as surrogates for propagule pressure and establishment risk in the community invasibility models. This HFI index aggregates major roadways, navigable waterways, railways, crop lands, pasture lands, the built environment, light pollution, and human population density. It therefore includes population data, disturbance (as anthropized lands), and human and goods exchanges measured over each watershed. As a complement to HFI we also considered the degree of river fragmentation (DOF) which measures the extent to which river networks are fragmented longitudinally by infrastructure, such as hydropower and irrigation dams (Grill et al. 2019). This anthropic disturbance is not redundant with the GDP or HFI, and has a potentially important effect on non-native species establishment (Su et al. 2021). We therefore consider that the GDP, HFI and DOF are the best available proxies of propagule pressure and establishment risk (see details in the methods lines 418-429).

Integrating the GDP, HFI and DOF as explanatory variables in our community invasibility model revealed a limited effect of propagule pressure, with these variables together having 22% relative influence in the model. It confirms that propagule pressure and environmental disturbances are influencing the invasibility of a community, but this effect remains limited compared to the differences of functional characteristics between recipient communities and non-native species, which account for about 60% of the variance explained by the model. Therefore, although we understand the questions raised about the role of propagule pressure on community invasibility, we here confirm its effect remains limited.

In the revised manuscript we therefore fully explained why we considered the GDP, HFI and DOF as proxies in the community invasibility models and clearly explain that failed introduction data is lacking, which will, we hope, respond to the queries of the reviewer. We also added a discussion about uncertainties about the use of such surrogates instead of a real

measure of propagule/colonization pressure and establishment rate in community invasibility models:

Lines 230-238: “Nevertheless, the three human-related variables collectively contributed 20.2% in explaining community invasibility, partially supporting the “human activity” hypothesis⁴ and verifying the significant impact of propagule pressure. However, since we used human activity as surrogates of propagule pressure, it remains difficult to discuss the true effect of propagule pressure. We thus highlight that the level of economic activity of a given watershed (expressed by the GDP) strongly affects the community invasibility through a possible increase in propagule pressure. In contrast, human footprint and the degree of river fragmentation had surprisingly small effects on community invasibility, whereas they are often considered to be the major predictors of the number of non-native species^{4,32,33}”

Considering species invasiveness about the number of introduction trials per species, again, refers to failed introductions that cannot be recorded because failed introduction attempts were rarely documented. Instead, human interest for each non-native species can be used as a proxy of frequency of introduction for each species. This has been used by Su et al. (2020), which considered that species of interest for humans have already been candidates for introduction. A more detailed way to consider introduction frequency would be to use the index of human affiliation/use. The index of human affiliation/use has been proposed by Blanchet et al. (2010) as the arithmetic average of the four categorical indexes of human use (fisheries, aquaculture, game fish and ornamental importance of each species) provided for each fish species in Fishbase. This index was used as an explanatory variable in the species invasiveness models to control for differences in introduction strength between species. Adding this human use index therefore provides information about the importance of propagule pressure in the models and thus tell us to which extent species of interest for humans and their associated traits play a role in species invasiveness. By controlling for this, the influence of functional traits in the species invasibility model will refer more to their ability to invade other regions than to their propensity to be chosen by humans.

Based on the results of the new model, we found that besides the functional traits, the human use index, used as a surrogate for propagule pressure in the model, is also a very important predictor (24.1%) of species invasiveness, suggesting that human interest is a non-negligible factor in the process of species invasion, especially in the introduction stage.

We therefore modified the MS to better consider human activities in our invasiveness models and considered the new results in the discussion:

Lines 403-410: “**Human use index:** Besides the functional traits of the fish species, to account for the impact of propagule pressure on species’ invasiveness, we applied the human use index as a surrogate in the models²⁵. FishBase⁵² provides four categorical indexes of human use, which refer to the fisheries, aquaculture, game fish and ornamental importance of each species. We assumed that each categorical index has an equivalent importance and assigned each index to 0 or 1 for each fish based on the description (e.g., 0 for never/rarely used, 1 for occasionally/commonly used). Then, we calculated the sum of the four indexes as the human use index, which varies between 0 (i.e. for species not used by human) and 4 (i.e. for species strongly used by human).”

Lines 126-128: “Partial dependency plots in Fig. 2 showed that fecundity contributed the most (29.2%) to species invasiveness, followed by the human use index (24.1%) and size-related traits (Size_PCA1, 17.5%).”

Lines 222-224: “..., our results show that the human use index, employed as a surrogate for propagule pressure in the model, remains an important predictor of species invasiveness, suggesting that human interest is a non-negligible factor in the process of species invasion, especially in the introduction stage².”

Beaury, E. M., Finn, J. T., Corbin, J. D., Barr, V. & Bradley, B. A. Biotic resistance to invasion is ubiquitous across ecosystems of the United States. *Ecology letters* **23**, 476-482 (2020).

Bernery, C. *et al.* Freshwater Fish Invasions: A Comprehensive Review. *Annual Review of Ecology, Evolution, and Systematics* **53**, 427-456 (2022).

Blanchet, S. *et al.* Non-native species disrupt the worldwide patterns of freshwater fish body size: implications for Bergmann’s rule. *Ecology Letters* **13**, 421-431 (2010).

Drake, D. A. R., Casas-Monroy, O., Koops, M. A. & Bailey, S. A. Propagule pressure in the presence of uncertainty: extending the utility of proxy variables with hierarchical models. *Methods in Ecology and Evolution* **6**, 1363-1371 (2015).

Essl, F. *et al.* Socioeconomic legacy yields an invasion debt. *Proceedings of the National Academy of Sciences* **108**, 203-207 (2011).

García-Berthou, E. The characteristics of invasive fishes: what has been learned so far? *Journal of Fish Biology* **71**, 33-55 (2007).

Grill, G. *et al.* Mapping the world’s free-flowing rivers. *Nature* **569**, 215 (2019).

Leprieur, F., Beauchard, O., Blanchet, S., Oberdorff, T. & Brosse, S. Fish invasions in the world's river systems: when natural processes are blurred by human activities. *Plos Biology* **6**, e28 (2008).

Lockwood, J. L., Hoopes, M. F. & Marchetti, M. P. *Invasion ecology*. 1st edn, (John Wiley & Sons, 2007).

Penone, C. *et al.* Imputation of missing data in life-history trait datasets: which approach performs the best? *Methods in Ecology and Evolution* **5**, 961-970 (2014).

Pyšek, P. *et al.* Disentangling the role of environmental and human pressures on biological invasions across Europe. *Proceedings of the National Academy of Sciences* **107**, 12157-12162 (2010).

Su, G., Villéger, S. & Brosse, S. Morphological sorting of introduced freshwater fish species within and between donor realms. *Global Ecology and Biogeography* **29**, 803-813 (2020).

Su, G. *et al.* Human impacts on global freshwater fish biodiversity. *Science* **371**, 835-838 (2021).

Reviewer #3 (Remarks to the Author):

General comments

A well written article that applies novel approaches to considering invasions and uncovers additional patterns of invasion using a large dataset across the continental US. The combination of analyses considering traits of invaders and invasibility of communities across the diversity of habitats and invaders provides compelling patterns related to life-history, community characteristics, etc. and sheds novel insights into how these two aspects of invasion (species-centered and (community-centered) are integrated. While the authors do a nice job of illustrating these two aspects separately, and attempt to bring them together in Figure 4 and the

discussion, an improved visual demonstrating how these patterns (individual vs community) are linked would be helpful to solidify how these pieces are integrated more broadly.

Thank you for your positive evaluation of our manuscript and for your constructive comments, which enhanced the quality of our manuscript.

Yes, the species invasiveness and community invasibility analyses were performed separately in our study. These two analyses answered the two main questions that species with what kinds of traits are more likely to invade other regions and communities with what functional attributes are easier to invade. Separating these two questions allowed us to consider the two processes independently, but we agree they can be linked, and that's why we also considered the functional correlation between native communities and invasive species in our models.

Going further in this direction, we could build on the present results to predict the probability that a given species will invade a focal community based on the species' functional traits as well as the attributes of the native community and their functional correlations. Linking both processes more tightly in the manuscript would add substantial complexity and length to the manuscript and blur the demonstration we made on the two separate questions, so we here preferred to keep the two questions separated. We nevertheless indicated in the conclusion that considering interaction between these two properties would be necessary in the future and added a theoretical figure (Fig. 5) to interpret how the functional traits can link the two aspects (lines 330-333):

“...we suggest that prediction of biological invasion should comprehensively consider the invasiveness of non-native species and the invasibility of recipient communities, as illustrated on Fig. 5, and encourage considering the interactions between those two properties in future studies.”

Fig. 5. Theoretical representation of the links between the non-native species invasiveness and recipient community invasibility. Non-native species are characterized by traits promoting invasiveness and by human use for each species as a proxy for propagule pressure. Combining species traits and propagule pressure predicts species invasiveness measured as the number of watersheds where the species established (shown in orange on the maps). Species A, B and C thus have a low, moderate and high invasiveness, respectively. Meanwhile, species C also has a high chance to establish in the three recipient communities because its functional traits locate the species close to the center of the functional space of the recipient communities (i.e., low centroid distance between non-native species and native assemblage) and the non-native species traits are not redundant with those of native species (i.e., high mean distance between native and non-native species). In contrast, species A and B either located near the boundaries the local functional space (high centroid distance, species A in Communities 1 and 2) or in crowded regions of the native functional space (low mean distance, species B in Community 1) will have low chances to establish, resulting in a low invasiveness for those species. Native community invasibility is therefore determined by high establishment probability regions in the functional space containing no or few native species (darker blue areas). cd: centroid distance; md: mean distance; FSpe: functional specialization.

Additionally, it would be helpful if the authors provided more insights into the relationships between species invasiveness and community invasibility and the predictors. Indeed, many of these relationships appear to illustrate patterns of thresholds, yet it is unclear if this is a data-driven pattern (and rug plots would help elucidate) or are these ecological relationships. If the latter, unpacking this in the discussion, even briefly, would help direct further studies and posit additional mechanisms that affect invasions. Finally, some specific comments are below.

We added rugs in Figs. 2,3. Examining rugs do not support thresholds due to data driven patterns, and thus suggests ecological relationships rather than data structure bias. We thus detailed the results on ecological patterns and briefly discussed possible mechanisms.

Lines 133-139: “However, none of these correlations are linear, e.g., species invasiveness does not change much at first until it increases dramatically when fecundity exceeds 1,000 eggs per female (point $\log_{10}FCt = 3$, Fig. 2) then stabilizes again over 100,000 eggs per female (point $\log_{10}FCt = 5$, Fig. 2). This broken-stick relationship is also found for body elongation, Temp_PCA1 and other traits. In contrast, size-related traits and trophic level showed a non-monotonic correlation to the invasiveness, with its highest values at medium-to-high values and then decreasing slightly thereafter (Fig. 2).”

Lines 205-208: “Notably, our results show that the relationship between species invasiveness and functional traits is non-linear, e.g., the broken-stick relationship in some traits, suggesting that invasiveness is more like a threshold function of traits and determined by joint effects of several traits.”

Specific comments

Line 90: Inconsistent use of "nonnative" as shown here, yet "non-native" elsewhere.

Corrected.

Lines 125-128: The response indicates more of a ~broken stick relationship between invasiveness and reproduction traits, body elongation, etc. suggesting more of thresholds than increasing relationships across the range of values.

We added more interpretation and discussion for the relationship (Lines 133-139, 205-208, see the response above).

Figure 3 and related text. The authors should also consider the form of these relationships. For example, why do these relationships asymptote and/or demonstrate a broken-stick relationship? Why does invasibility not continue to decline when cd levels are >10 ? Why does invasibility not continue to increase when Area $>\sim 18,000$? Are there thresholds by which invasibility changes dramatically? While I understand this is a short-format journal, these results have important implication about which communities are the most vulnerable

We developed this in the results and discussion.

Lines: 165-170: “All correlations between community invasibility and predictors were non-linear monotonically decreasing or increasing relationships. For instance, community invasibility was strongly negatively related to centroid distance between native and non-native species (cd) values lower than a threshold (0.12), and then not sensitive to the highest range of cd values. In the same way, community invasibility increased with watershed area up to 12,000 km^2 , but then stabilized for larger areas (Fig. 3).”

Lines 256-261: “Our findings also tend to support the biotic acceptance hypothesis, which is verified by the positive correlation between the watershed area (and thus the native species richness), NPP and invasibility, therefore confirming that species-rich native assemblages are not insured against non-native species establishment. This was verified for watershed areas up to 12,000 km², whereas for larger watersheds, areas are no longer linked to invasibility, which stays maximal.”

Lines 278-281: “This also explains the low and constant invasibility for higher centroid distance (*cd*) values (above a threshold of 0.12) due to non-native species holding traits falling out of the range of traits of the native community (see species A in Fig. 5).”

Also, note the panel for Area has 6000 instead of 60000.

Changed. We checked our data and removed a few watersheds covering the Great Lakes (watersheds) of Great Lakes in the border, so the maximum area reduced to about 24,000 km². This change did not affect the final results.

Lines 168-169: Given the correlations between streamflow and productivity, among other factors, this statement should be refined.

We agree with your comment, and removed this sentence.

Lines 203-204: This statement should be tempered. For example, *cd* explained 16.8% of the variation in invasibility, yet FPT (9.1%) together with DOF (6.2%) explain 15.3%. At the same time, landscape attributes (area, NPP), together explain 23.2%. Certainly, all are having an effect, but the language should be tempered to reflect the data.

We modified the related texts and added more discussion about human influence on community invasibility (lines 230-238):

“Nevertheless, the three human-related variables collectively contributed 22.1% in explaining community invasibility, partially supporting the “human activity” hypothesis⁴ and verifying the notable impact of propagule pressure. However, since we used human activity surrogates of propagule pressure, it remains difficult to discuss the true effect of propagule pressure. We thus highlight that the level of economic activity of a given watershed (expressed by the GDP) strongly affects the community invasibility through a possible increase in propagule pressure. In contrast, human footprint and the degree of river fragmentation had surprisingly small effects on community invasibility, whereas they are often considered to be the major predictors of the number of non-native species^{4,32,33}”

Lines 210-213: The inconsistency may also reflect the lack of propagule pressure information considered herein. See comments in methods as well, which question the links between FPT and propagule pressure as considered herein.

We added more discussion on this inconsistency (lines 241-248) and we also gave more explanation about the links between human-related variables and propagule pressure, please also refer to our answer to the previous point.

“Our regional findings contrast with these global results as we report a relatively weak influence of human activities on community invasibility patterns across the US watersheds. This inconsistency might be rooted in the different spatial scales considered. The watershed spatial scale used here may have highlighted the role of species interactions within the communities on invasibility at the expense of some human disturbances that might be better assessed over

larger spatial scales. Besides, at the watershed scale, the measurement of river fragmentation only accounts for dams located in the focal watershed, while hydrologically mediated effects might spread far downstream^{34,35}.”

Lines 231-232: The statement that non-natives are "avoiding" competition is not necessarily supported by the data. There are hints at this, but the text concluding competition is a bit preliminary-especially given the challenges of demonstrating competition in the wild.

We agree. We changed the text to a possible reduction of competitive interactions, and underlined this process cannot be tested with our data (lines 271-276):

“This indicates consistency in the pattern of most non-native species packing into the center of the native species’ functional space, but keep distance from their native neighbors, which might avoid competitive interactions that are prone to reduce the chances of establishment. Although competitive interactions cannot be properly tested here, field studies on specific communities to test this hypothesis are warranted.”

Also, despite the patterns in the inset boxes around the perimeter in Figure 4, it would be helpful to demonstrate the consistency of these patterns across watersheds to help support the text here—especially since this is a major outcome of this analysis. The authors speak to an occurrence or two and while these inset boxes do help, considering a visual (even in the Supplemental) to support the patterns between cd and md as they pertain to the ecology of invasions more broadly would strengthen the text here.

We added more explanations about this part and Fig. 4 (lines 267-274). We also added a visual (Fig. 5) to illustrate the patterns, please see our response to your general comment for more detail on this figure, and references to this figure in the discussion.

“For instance, with similar native species number, the fish communities in watersheds with less invasive species tended to have a higher centroid distance and a lower mean distance, e.g. Caddo lake, Waiska (Fig. 4). In contrast, the communities in Upper Columbia-Entiat, San Pablo Bay, Middle Guadalupe, and Mohawk with more invasive species tended to have a lower centroid distance and a higher mean distance (Fig. 4). This indicates consistency in the pattern of most non-native species packing into the center of the native species’ functional space but keeping distance from their native neighbors, which might avoid competitive interactions that are prone to reduce the chances of establishment.”

Lines 278-290: Given that occurrence can be affected by sampling intensities, etc., it would be helpful if the authors provided (1) a measure of how much sampling occurred in each watershed (e.g., mean, SD across watersheds or something similar); and 2) what is the temporal dataset that was included in these data? I.e., what range of dates of sampling and occurrence were included?

We here used well established databases on fish occurrences in US watersheds. NatureServe (<https://www.natureserve.org>) and USGS (<http://nas.er.usgs.gov>). There is no mention of sampling effort in these databases, but they gather cumulative observations since 1996 and represent the most comprehensive information on species records per watershed. This would indeed be a sampling effort bias problem if we were focusing on abundances but we here only consider occurrences, and thus just need species lists per watershed.

Although less sensitive to sampling effort than abundances, we cannot exclude incompleteness in species lists, but given that the two data sources are commonly used in the scientific literature

based on fish occurrence data (e.g., Muneeppeerakul et al. 2008, Anas & Mandrak 2021, Comte et al. 2021, Qian et al. 2021), and recognized in the literature to provide the most comprehensive occurrence data of native and non-native fish species in the US, we are quite confident in the relevance of the data.

This is now indicated in the methods (lines 349-351)

Anas, M. U. M. & Mandrak, N. E. Drivers of native and non-native freshwater fish richness across North America: Disentangling the roles of environmental, historical and anthropogenic factors. *Global Ecology and Biogeography* **30**, 1232-1244 (2021).

Comte, L., Grantham, T. & Ruhi, A. Human stabilization of river flows is linked with fish invasions across the USA. *Global Ecology and Biogeography* **30**, 725-737 (2021).

Muneeppeerakul, R. *et al.* Neutral metacommunity models predict fish diversity patterns in Mississippi–Missouri basin. *Nature* **453**, 220-222 (2008).

Qian, H. *et al.* Taxonomic and phylogenetic β -diversity of freshwater fish assemblages in relationship to geographical and climatic determinants in North America. *Global Ecology and Biogeography* (2021).

Also, to broaden the inference to provide the readers with the range of abiotic conditions considered herein, it would be helpful to add a summary table to the Supplemental.

We added a supplementary table (Table S3) to show the statistical summary of the abiotic variables. In addition, we also provided all the used abiotic data per watershed in the data and code availability section in our MS.

Lines 313-314: On lines 295-297 the authors present a total of 862 species considered herein. Why the discrepancy?

959 species with trait data were initially considered but only 859 species (in the revised version) occur in the considered watersheds as natives or established non-natives. We modified the sentence to keep only the 859 species in the MS to avoid confusion.

Lines 326-330: 20% is a high for missing data and despite the high correlations from the missForest analyses, it is hard to grasp if the taxa with missing values were those that were well outside the range of values of fishes with data. Were there any patterns as to specific taxonomic groups of invaders or native fishes without data? I.e., were these missing species closely related to those species where data occurred?

Random forest is a very powerful method and one of the best methods to fill missing values in the trait matrix, even without phylogenetic information (Penone et al 2014). Although adding phylogenetic information can improve the model performance, ‘missForest’ is not mainly based on the phylogenetic or taxonomic information to fill missing values. Thus, RF is not overly sensitive to the distribution of missing values over taxonomic groups. Nonetheless, we checked the distributions of missing data for natives, nonnatives and each taxonomic order in our dataset (figure below, the number above the bars indicates the number of species in each group). We can see that only two small orders (accounting each for 2 species and no non-native species in those orders) have more than 50% missing values. For native/nonnative groups and most of the species rich orders, the percentage of missing values is around 20%, which means the missing values distribute quite evenly in our dataset. Moreover, the percentage of missing values for non-native species is 12.1%, indicating that non-native species are fairly well functionally described. We added this figure as Fig. S6 in the supplementary.

Figure S6. Distribution of missing values among the entire dataset, native, non-native, and taxonomic orders. Values above the bars represent the numbers of species in the groups.

Penone, C. *et al.* Imputation of missing data in life-history trait datasets: which approach performs the best? *Methods in Ecology and Evolution* **5**, 961-970 (2014).

Line 351: In the previous section (~338-348), the authors present 6 functional metrics yet here consider 5-dimensional functional space?

We apologize for the lack of clarity with respect to the functional space we consider. Here, the 5-dimensional functional space is built from the first five principal coordinate axes in the PcoA. This 5-dimensional space was used to calculate 6 functional metrics (FRic, FSpe, FOr, FDis, FDiv and FEve) that represent the size, the position of communities in the functional space as well as the distribution of species in the functional space. Those metrics are therefore independent from the number of dimensions considered in the functional space. We rephrased these sentences to make it clearer (lines 415-424):

“We then used principal coordinate analysis (PCoA) to build the functional space on the first five principal coordinate axes, giving rise to a five-dimensional functional space which explained over 80% of the total variance. We removed watersheds with fewer than six species to meet the criteria for calculating functional diversity indices, which resulted in 1,868 watersheds for the following analyses. Then we computed six complementary functional indices that are frequently used in functional diversity studies^{19,57-59}: functional richness (FRic), functional evenness (FEve), functional divergence (FDiv), functional dispersion (FDis), functional specialization (FSpe), and functional originality (FOri). These six metrics were used to represent the functional size and structure (measured on the five-dimensional functional space) of the recipient fish community in each watershed.”

Line 367: Area is not necessarily a measure of habitat diversity and the relationship with species richness is likely driven by species~area relationships.

We agree that the species-area relationship is more accurate. As discussed in Oberdorff et al (2011), three main nonexclusive mechanisms can explain the species-area relationship: (1) the size-dependent extinction rate, (2) the size-dependent speciation rate, and (3) the diversity of the habitat. Thus, many studies considered river basin areas as a surrogate for fish habitat size and diversity (Leprieur et al 2008, Oberdorff et al 2019). We changed “habitat diversity” to “habitat size and diversity” to make it clearer.

Leprieur, F., Beauchard, O., Blanchet, S., Oberdorff, T. & Brosse, S. Fish invasions in the world's river systems: when natural processes are blurred by human activities. *Plos Biology* **6**, e28 (2008).

Oberdorff, T. *et al.* Global and regional patterns in riverine fish species richness: a review. *International Journal of Ecology* **2011**, 967631 (2011).

Oberdorff, T. *et al.* Unexpected fish diversity gradients in the Amazon basin. *Science advances* **5**, eaav8681 (2019).

Lines 366-368: Are there relationships linking any of these metrics in FPT with propagule pressure? Given the importance of propagule pressure on invasive species establishment, understanding how well this is captured in this dataset is important. If not well captured, or understood, acknowledging this may be needed. I understand this is a surrogate, but not clear how well this aspect is captured in FPT.

I can think of numerous examples where, for example, trains in the 1800s were used to transport non-native fishes across the US--which likely wouldn't be captured in these metrics for propagule pressure.

The lack of data about propagule pressure, including therefore introduction attempts and failed introductions, apply to most aquatic ecosystems as frequently underlined in the literature (e.g., Garcia-Berthou 2007, Drake et al. 2015; Bernery et al. 2022). The data considered in our paper, although being among the most detailed regional data on fish invasion does not break this rule as it only contains 1,200 records of failed introduction over a total of more than 13,000 introduction events. It would therefore not be realistic to use the sum of failed and successful introduction events as a measure of propagule pressure. We therefore cannot link propagule pressure to HFI (formerly abbreviated FTP) in our data.

Despite this, studies dealing with invasions usually consider human disturbances on recipient ecosystems and human interest for non-native candidate species as relevant proxies of propagule pressure (Pyšek et al. 2010).

Human disturbances on recipient ecosystems are often approximated using the human footprint index (HFI), the Gross Domestic Product (GDP) or the human population density, because increasing the number of humans and economic exchanges increases the chances of introduction (Leprieur et al. 2008; Essl et al. 2011; Beaury et al. 2020, Comte et al. 2021). In addition, increasing human population and economic activity also increases environmental disturbances leading to the decline of some sensitive native species, and open niches for non-native species that are often opportunistic. This explains why the HFI, GDP has frequently been used as a proxy of propagule pressure (Leprieur et al. 2008, Essl et al. 2011; Beaury et al. 2020).

We therefore used the GDP and HFI as surrogates to propagule pressure in the community invasibility models. This HFI index aggregates major roadways, navigable waterways, railways, crop lands, pasture lands, the built environment, light pollution, and human population density. It therefore includes population data, disturbance (as anthropized lands), and human and goods

exchanges measured over each watershed. As a complement to HFI we also considered the degree of river fragmentation (DOF) which measures the degree to which river networks are fragmented longitudinally by infrastructure, such as hydropower and irrigation dams (Grill et al. 2019). This anthropic disturbance is not redundant with the GDP or HFI, and has a potential important effect on non-native species establishment (Su et al. 2021). We therefore consider that the GDP, HFI and DOF are the best available proxies of propagule pressure and establishment risk (see lines 445-456 quoted below).

Even if these measures do not account for ancient introductions (e.g., 19th century introduction through railways, as you noticed), the early development of railways anticipated the population increase and economic development of a region, so this information is already considered in the HFI. More generally, ancient introductions were scarce, and the rapid increase in non-native species introduction rates in the last decades has often been demonstrated, making such early introduction events quite rare compared to the more recent ones (Vander Zanden 2005; Seebens et al. 2017).

In the revised manuscript we therefore fully explained why we considered the GDP, HFI and DOF as proxies in the community invasibility models and clearly explain that failed introduction data is lacking (lines 424-435). We also added a discussion about uncertainties about the use of such surrogates instead of a real measure of propagule pressure in community invasibility models (lines 2230-238).

Lines 445-456: “Because data on propagule pressure is missing for most species^{23,61}, it is often assessed using various proxies relating to human activities^{4,48,62}. We therefore used the Gross Domestic Product (GDP) and human footprint index (HFI) as surrogates for propagule pressure and establishment risk in the community invasibility models. The HFI aggregates major roadways, navigable waterways, railways, crop lands, pasture lands, the built environment, light pollution, and human population density. It therefore includes population data, disturbance (as anthropized lands), and human and goods exchanges measured over each watershed. As a complement to GDP and HFI, we also considered the degree of river fragmentation (DOF) which measures the degree to which river networks are fragmented longitudinally by infrastructure, such as hydropower and irrigation dams¹. This anthropic disturbance is not redundant with the GDP or HFI, and has a potential important effect on non-native species establishment². We therefore considered that the GDP, HFI and DOF are the best available surrogates for propagule pressure and establishment risk.”

Lines 230-238: “Nevertheless, the three human-related variables collectively contributed 22.1% in explaining community invasibility, partially supporting the “human activity” hypothesis⁴ and verifying the notable impact of propagule pressure. However, since we used human activity surrogates of propagule pressure, it remains difficult to discuss the true effect of propagule pressure. We thus highlight that the level of economic activity of a given watershed (expressed by the GDP) strongly affects the community invasibility through a possible increase in propagule pressure. In contrast, human footprint and the degree of river fragmentation had surprisingly small effects on community invasibility, whereas they are often considered to be the major predictors of the number of non-native species^{4,32,33}”

Beaury, E. M., Finn, J. T., Corbin, J. D., Barr, V. & Bradley, B. A. Biotic resistance to invasion is ubiquitous across ecosystems of the United States. *Ecology letters* **23**, 476-482 (2020).

Bernery, C. *et al.* Freshwater Fish Invasions: A Comprehensive Review. *Annual Review of Ecology, Evolution, and Systematics* **53**, 427-456 (2022).

- Drake, D. A. R., Casas-Monroy, O., Koops, M. A. & Bailey, S. A. Propagule pressure in the presence of uncertainty: extending the utility of proxy variables with hierarchical models. *Methods in Ecology and Evolution* **6**, 1363-1371 (2015).
- Essl, F. *et al.* Socioeconomic legacy yields an invasion debt. *Proceedings of the National Academy of Sciences* **108**, 203-207 (2011).
- García-Berthou, E. The characteristics of invasive fishes: what has been learned so far? *Journal of Fish Biology* **71**, 33-55 (2007).
- Grill, G. *et al.* Mapping the world's free-flowing rivers. *Nature* **569**, 215 (2019).
- Leprieur, F., Beauchard, O., Blanchet, S., Oberdorff, T. & Brosse, S. Fish invasions in the world's river systems: when natural processes are blurred by human activities. *Plos Biology* **6**, e28 (2008).
- Penone, C. *et al.* Imputation of missing data in life-history trait datasets: which approach performs the best? *Methods in Ecology and Evolution* **5**, 961-970 (2014).
- Pyšek, P. *et al.* Disentangling the role of environmental and human pressures on biological invasions across Europe. *Proceedings of the National Academy of Sciences* **107**, 12157-12162 (2010).
- Seebens, H. *et al.* No saturation in the accumulation of alien species worldwide. *Nature communications* **8**, 14435 (2017).
- Su, G., Villéger, S. & Brosse, S. Morphological sorting of introduced freshwater fish species within and between donor realms. *Global Ecology and Biogeography* **29**, 803-813 (2020).
- Vander Zanden, M. J. The success of animal invaders. *Proceedings of the National Academy of Sciences* **102**, 7055-7056 (2005).

Reviewer comments, second round –

Reviewer #1 (Remarks to the Author):

I was Reviewer #1 of the previous manuscript (MS). I made many suggestions to that MS and they have been carefully considered in the current manuscript, which has improved considerably in my opinion. As the authors acknowledge, there are some caveats in the analyses and limitations of the data but these are unavoidable in this type of manuscripts.

1) The new Fig. 5 is helpful to summarize the main point made in the manuscript (i.e. "Community invasibility peaked when the functional distance among native species was high, leaving unoccupied functional space for the establishment of potential invaders." (Abstract)). However, I would have preferred that it be based on real examples (as Fig. 4) more than a "theoretical representation". The authors should be able to provide such an example (perhaps building on Fig. 4).

2) L. 136. I think I would not use the term "broken-stick relationship", which does not seem frequent in the ecological literature, can be confounded with broken-stick models (e.g. <https://doi.org/10.2307/3545108>), and implies that a segmented or piecewise regression might be appropriate (which has not been analyzed in the paper).

3) After rereading the revised MS, I am not a bit confused about the methods. How was community invasibility (i.e. the response variable in Fig. 3) exactly measured? I do not see this well explained in the methods of the MS (?). If it is "The established non-native species number in each community" (L. 49)?, is the community the fish found in the HUC8 watershed? What is the rationale (or previous literature support) of defining community invasibility this way? The number of non-native species seems more an invasion rate more than "invasibility". If there are many nonnative species or not (e.g. by degradation) should also affect community invasibility.

4) I think the methods and rationale of the two distances need to be better explained: What is the meaning or purpose of square brackets in Fig. 1? If $CX = [CX_i]$, why to use both? (Minor point: you use italics for some symbols in the Fig. but not in the caption).

5) If "md is the mean distance between all non-native species and their nearest native", why not to also consider the distance among nonnative species as well? If a nonnative species is introduced to a place where another very similar nonnative (e.g. same genus) is already established, this should clearly influence community invasibility.

6) Related to my comments in the previous version, the negative relationship between community invasibility (CI) and cd (centroid distance) seems a mathematical necessity. If CI is the number of non-native species, it is likely to influence cd (more species imply less distance simply by chance). cd would be a consequence of "CI" more than a cause or predictor.

Reviewer #3 (Remarks to the Author):

General comments

The authors have done a nice job of rectifying comments and additional text/figures do a good job of explaining the results. With novel approaches and an expansive analysis of invasion/invasibility, the manuscript provides new clarity to invasions across the US that should also provide insights beyond the focal area. I only have a few remaining comments.

Specific comments

Lines 87-88: "factor" needs to be changed to plural and/or the sentence needs revising.

Lines 85-91: Does it always have to be one or the other? Propagule pressure can also allow for

success, simply as a function of overcoming poor abiotic (e.g., climatic) conditions during introduction in one or a few events. Thus, propagule pressure over time can help overcome these periods (e.g., Lockwood et al. 2005, Trends in Ecology and Evolution) and conversely do well during "windows of opportunity". One could argue that the invasion success could be exacerbated by the patterns of functional traits where propagule pressure is high, etc.

Lines 216: At some point, the authors should clarify that the term "invade" here, includes both intentional and unintentional invasions, where the latter is largely a function of secondary spread/radiation across connected riverscapes. The former, which is much more common here in the example of bluegill and carp, is predominantly from stocking of sportfish.

Lines 239-253: The authors should consider, even briefly that human disturbance may indirectly affect invasibility as human effects may lead to declines in native species which may foster invasion. This may be difficult to identify with occurrence data as decimated populations of native species may still "occupy" an area, but at very low relative numbers.

Lines 261-262: The term "unveil the mechanism" seems a bit strong here. Mechanisms are nearly impossible to identify with these types of analyses.

Lines 403-405: The order of this sentence is odd. Consider reorganizing, to "Besides...the fish, we applied the.....in the models to account for the impact of propagule pressure on ...".

Lines 425-429: As is the text of definitions is awkward as a stand-alone paragraph. Consider moving text to line 422 before "these". And add ~"The indices are briefly defined as the following:".

Lines 445-456: The location of this text describing propagule pressure is odd. It sits between the overview of the factors considered in the biotic acceptance/resistance hypotheses and the more detailed descriptions of each of the factors.

Lines 495-498: The logic does not follow here. Paraphrased, this sentence reads as "Since we aim to explore general patterns, a few species are not adjusted to the pattern". I.e., the extreme trait values are not due to the authors' goal to explore patterns.

REVIEWER COMMENTS

Reviewer #1 (Remarks to the Author):

I was Reviewer #1 of the previous manuscript (MS). I made many suggestions to that MS and they have been carefully considered in the current manuscript, which has improved considerably in my opinion. As the authors acknowledge, there are some caveats in the analyses and limitations of the data but these are unavoidable in this type of manuscripts.

1) The new Fig. 5 is helpful to summarize the main point made in the manuscript (i.e. "Community invasibility peaked when the functional distance among native species was high, leaving unoccupied functional space for the establishment of potential invaders." (Abstract)). However, I would have preferred that it be based on real examples (as Fig. 4) more than a "theoretical representation". The authors should be able to provide such an example (perhaps building on Fig. 4).

The panels of Fig. 4 provide concrete examples of the theoretical representation shown in Fig. 5, and centroids for native and non-native species were added to these functional spaces to ease interpretation.

Replacing the theoretical Fig. 5 by real case examples would be difficult because failed introduction records are lacking, making it difficult to illustrate the link between species invasiveness and community invasibility. We therefore preferred to keep theoretical examples on the Fig. 5 to provide a synthetic representation of the main findings, but in the main text, we refer to the case examples from Fig. 4 to illustrate the processes shown in Fig. 5.

Moreover, in the Fig. 5 we used real species examples to illustrate functional traits and invasion patterns. The three represented species are *Thymallus arcticus*, *Sander vitreus* and *Cyprinus carpio*. Therefore, only the functional space representation remains theoretical because failed introduction attempts are not available, and representing such hypothetical failed attempts on real sites would be misleading.

2) L. 136. I think I would not use the term "broken-stick relationship", which does not seem frequent in the ecological literature, can be confounded with broken-stick models (e.g. <https://doi.org/10.2307/3545108>), and implies that a segmented or piecewise regression might be appropriate (which has not been analyzed in the paper).

You are right, we removed the term "broken-stick relationship" to avoid confusion and now simply describe a steep increase followed by a stabilization (lines 157-158).

3) After rereading the revised MS, I am not a bit confused about the methods. How was community invasibility (i.e. the response variable in Fig. 3) exactly measured? I do not see this well explained in the methods of the MS (?). If it is "The established non-native species number in each community" (L. 49)?, is the community the fish found in the HUC8 watershed? What is the rationale (or previous literature support) of defining community invasibility this way? The number of non-native species seems more an

invasion rate more than "invasibility". If there are many nonnative species or not (e.g. by degradation) should also affect community invasibility.

Sorry for the confusion. Yes, here we used the established non-native species number in each community (ie, the fish assemblage in each HUC8 watershed) as a proxy of community invasibility. We modified the sentence to make it clearer (Lines 55-57): "...here measured as the established non-native species number in each community (i.e., the fish assemblage in each HUC8 watershed)"

This was also clarified in the methods section (Lines 592-593): "We then computed the established non-native species number in each of the 1,868 watersheds as a proxy for community invasibility,..."

The rationale for defining community invasibility was also added to the main text, by refereeing to previous literature support such as Elton (1958) who defined invasibility in his seminal book titled "The ecology of invasions by animals and plants" as the number of non-native species encountered in a territory. The same definition was used by Levine (2013) in the *Encyclopedia of Biodiversity* (Second Edition) where invasibility was defined as "the actual number or proportion of alien species in a community, habitat, or region"

Many other studies have also used the number of exotic species as an indicator of invasibility (e.g., Elton 1958, Lonsdale et al. 1999, Fridley et al. 2004, Herben et al. 2004, Lockwood 2009, Levin 2013), and we now refer to these articles in the manuscript (lines 55-57).

- Elton, C. S. (1958). *The ecology of invasions by animals and plants*: Springer Nature.
- Fridley, J. D., Brown, R. L., & Bruno, J. F. J. E. (2004). Null models of exotic invasion and scale-dependent patterns of native and exotic species richness. *Ecology*, 85(12), 3215-3222.
- Herben, T., Mandák, B., Bímová, K., & Münzbergová, Z. J. E. (2004). Invasibility and species richness of a community: a neutral model and a survey of published data. *Ecology*, 85(12), 3223-3233.
- Levin, S. A. (2013). *Encyclopedia of biodiversity*: Elsevier Inc.
- Lockwood, J. L., Cassey, P., & Blackburn, T. M. (2009). The more you introduce the more you get: the role of colonization pressure and propagule pressure in invasion ecology. *Diversity and Distributions*, 15(5), 904-910. doi:10.1111/j.1472-4642.2009.00594.x
- Lonsdale, W. M. J. E. (1999). Global patterns of plant invasions and the concept of invasibility. *Ecology*, 80(5), 1522-1536.

4) I think the methods and rationale of the two distances need to be better explained: What is the meaning or purpose of square brackets in Fig. 1? If $CX = [CX_i]$, why to use both? (Minor point: you use italics for some symbols in the Fig. but not in the caption).

We detailed the figure caption to better explain the methods and rationale of the two distance metrics.

The square brackets in Fig. 1 represent the coordinates of the centroids. For instance if we have n traits: 1, 2, ..., n, then $C_X = [C_{X_i}] = [C_{X_1}, C_{X_2}, \dots, C_{X_n}]$. Here we introduced

$[C_{Xi}]$ and $[C_{Yi}]$ to better show how the cd is calculated. This is now clarified in the figure caption.

We also unified the italics for all the symbols in the figure and caption.

The revised caption is as follows:

Fig. 1. An example showing how centroid distance (cd) and mean distance (md) are computed. The p and q individual native and non-native species in a n -dimensional trait space (here $n = 2$) are represented by blue and red circles. Vector Y_j represents the position of non-native species j and vector X_j is the position of its nearest native species. d_j is the distance between X_j and Y_j . md is the mean distance between all non-native species and their nearest native neighbors. C_X and C_Y (triangles) are the centroids of the p native species and q non-native species, with $[C_{Xi}]$ and $[C_{Yi}]$ representing the coordinates of the centroids according to all traits, i.e. $[C_{X1}, C_{X2}, \dots, C_{Xn}]$, $[C_{Y1}, C_{Y2}, \dots, C_{Yn}]$. Here, $C_X = [C_{Xi}]$ and $C_Y = [C_{Yi}]$, where C_{Xi} and C_{Yi} are the mean value of trait i for native and non-native species. cd is the distance between the centroids for native (C_X) and non-native (C_Y) species.

5) If "md is the mean distance between all non-native species and their nearest native", why not to also consider the distance among nonnative species as well? If a nonnative species is introduced to a place where another very similar nonnative (e.g. same genus) is already established, this should clearly influence community invasibility.

We agree that already established species do influence community invasibility. Therefore, it would be useful to consider the distance between each non-native species and the rest of the community (including natives and non-natives), but considering non-natives one by one would also require considering the temporal dynamics of invasion in each watershed, which is unfortunately impossible because of the lack of temporal data on non-native species settlement for most watersheds. We therefore considered the nonnative species altogether to compute community invasibility. This is now indicated in the manuscript (lines 505-510).

It would nevertheless be nice to consider changes in md including native and non-native species through the invasion process. This would require recalculating the functional space after each non-native species establishment and quantifying how md changes through time, but this would have to be another research project focused only on the small subset of watersheds where the invasion sequence is fully known.

6) Related to my comments in the previous version, the negative relationship between community invasibility (CI) and cd (centroid distance) seems a mathematical necessity. If CI is the number of non-native species, it is likely to influence cd (more species imply less distance simply by chance). cd would be a consequence of "CI" more than a cause or predictor.

Yes, there should be a mathematical link between CI and cd if native and non-native species were randomly distributed in the functional space, and then, we agree that cd would be a consequence of CI. However, the real situation is markedly different, as shown in the supplementary Fig. S1-S2, most traits values of invasive species are not randomly distributed in the functional space, and tend to cluster in certain regions (while generally

keeping more functional distance between each other than between their nearest native neighbour). This is also illustrated in the case examples shown on the panels of Fig.4. In addition, relative locations of invasive species in the functional space of native communities does affect their invasive performance and thus influence the community invasibility, as demonstrated by Elleouet et al. (2014) for marine fish assemblages and by Catford et al. (2019) for grasslands. Thus, despite a mathematical link between CI and *cd*, the non-randomness and the clustering of functional traits of non-native species in areas of the functional space unoccupied by native species makes *cd* more a predictor than a consequence of community invasibility.

This is now explained as follows in the manuscript: “Although there would be mathematical link between community invasibility and *cd* if native and non-native species were randomly distributed in the functional space, most traits values of invasive species are not randomly distributed in the functional space, and tend to cluster in certain regions (Figs S1 & S2, see also case examples on Fig.4). Thus, despite a potential mathematical link between CI and *cd*, the non-randomness and the clustering of functional traits of non-native species in areas of the functional space unoccupied by native species makes *cd* a main predictor of community invasibility.” (Lines 183-190).

Catford, J. A., et al. (2019). "Traits linked with species invasiveness and community invasibility vary with time, stage and indicator of invasion in a long-term grassland experiment." *Ecology Letters* **22**(4): 593-604.

Elleouet, J., et al. (2014). "A trait-based approach for assessing and mapping niche overlap between native and exotic species: the Mediterranean coastal fish fauna as a case study." *Diversity and Distributions* **20**(11): 1333-1344.

Reviewer #3 (Remarks to the Author):

General comments

The authors have done a nice job of rectifying comments and additional text/figures do a good job of explaining the results. With novel approaches and an expansive analysis of invasion/invasibility, the manuscript provides new clarity to invasions across the US that should also provide insights beyond the focal area. I only have a few remaining comments.

Thank you for your positive evaluation.

Specific comments

Lines 87-88: "factor" needs to be changed to plural and/or the sentence needs revising. Changed.

Lines 87-88: “Instead, if functional traits and structure are more important factors, the influence of propagule pressure will be lower.”

Lines 85-91: Does it always have to be one or the other? Propagule pressure can also allow for success, simply as a function of overcoming poor abiotic (e.g., climatic) conditions during introduction in one or a few events. Thus, propagule pressure over time

can help overcome these periods (e.g., Lockwood et al. 2005, Trends in Ecology and Evolution) and conversely do well during "windows of opportunity". One could argue that the invasion success could be exacerbated by the patterns of functional traits where propagule pressure is high, etc.

We agree that propagule pressure and functional traits can both affect the invasion success in some particular cases. This has been indicated in the introduction as follows:

“For instance, if propagule pressure plays a major role, we should not detect a strong pattern of functional traits and structure between native and invasive species, although invasion success can, in some situations, be exacerbated by the pattern of functional traits when propagule pressure is high [e.g., propagule pressure can ease invasion by providing species with traits overcoming unusual conditions (e.g., climatic) in one or a few events]”. (Lines 97-102)

Lines 216: At some point, the authors should clarify that the term "invade" here, includes both intentional and unintentional invasions, where the latter is largely a function of secondary spread/radiation across connected riverscapes. The former, which is much more common here in the example of bluegill and carp, is predominantly from stocking of sportfish.

Yes, we agree. We replaced “invade” to “establish in”, which is more accurate in our case. We also replaced “invaders” by “non-native established species” in several places in the manuscript.

Lines 239-253: The authors should consider, even briefly that human disturbance may indirectly affect invasibility as human effects may lead to declines in native species which may foster invasion. This may be difficult to identify with occurrence data as decimated populations of native species may still "occupy" an area, but at very low relative numbers.

We added a sentence in the manuscript to illustrate the indirect effect of human disturbance

Lines 534-536:

“In addition, these human activities may also indirectly affect invasibility by reducing the abundance or number of native species, and thus reduce biotic resistance of the native fauna and ease non-native species establishment.”

Lines 261-262: The term "unveil the mechanism" seems a bit strong here. Mechanisms are nearly impossible to identify with these types of analyses.

We rephased the sentence (Lines 305-306): “..., we were able to shed further light on the mechanisms of community invasibility.”

Lines 403-405: The order of this sentence is odd. Consider reorganizing, to "Besides...the fish, we applied the.....in the models to account for the impact of propagule pressure on ...".

Reorganized.

Lines 466-468: “Besides the functional traits of the fish species, we applied the human use index as a surrogate in the models to account for the impact of propagule pressure on species’ invasiveness.”

Lines 425-429: As is the text of definitions is awkward as a stand-alone paragraph. Consider moving text to line 422 before "these". And add ~"The indices are briefly defined as the following:".

Reorganized. See lines 487-493

Lines 445-456: The location of this text describing propagule pressure is odd. It sits between the overview of the factors considered in the biotic acceptance/resistance hypotheses and the more detailed descriptions of each of the factors.

We moved this text below the text describing environmental factors.

See lines 522-538.

Lines 495-498: The logic does not follow here. Paraphrased, this sentence reads as "Since we aim to explore general patterns, a few species are not adjusted to the pattern". I.e., the extreme trait values are not due to the authors' goal to explore patterns.

We rephrased this sentence to make it logical.

Lines 577-583: “In addition, a few species (seven out of 300 species) appeared as outliers due to extreme traits values. These traits might reflect evolutionary contingency or erroneous traits measurements, and do not follow the general patterns of the relationship between species invasiveness and fish functional traits. Those species were removed from the analysis to avoid an undue influence of those outliers, make the pattern clearer and optimize results visualization.”

Reviewer comments, third round –

Reviewer #1 (Remarks to the Author):

I was Reviewer #1 of the previous manuscript (MS). I now fully read the response and revised manuscript. In my opinion, the authors have carefully considered the many previous suggestions.

Minor typo: replace "gc/m2/yr" with "g C m-2 year-1" in L. 519

Emili García-Berthou